# Identifying grain boundary and intragranular pinning centres in Sm₂(Co,Fe,Cu,Zr)₁₇ permanent magnets to guide performance optimisation

Stefan Giron[1], Nikita Polin[2], Esmaeil Adabifiroozjaei [3], Yangyiwei Yang [1], Fernando Maccari [1], András Kovács [4], Trevor P. Almeida [5], Dominik Ohmer[1,6], Kaan Üstüner[6], Alaukik Saxena[2], Matthias Katter[6], Iliya A. Radulov [1], Christoph Freysoldt [2], Rafal E. Dunin-Borkowski [4], Michael Farle [7], Karsten Durst[1], Hongbin Zhang [1], Lambert Alff [1], Katharina Ollefs [7], Bai-Xiang Xu [1], Oliver Gutfleisch[1,2], Leopoldo Molina-Luna [3] ✉, Baptiste Gault [2,8,9] ✉ & Konstantin P. Skokov [1] ✉

Permanent magnets draw their properties from a complex interplay of chemical composition and phase, each with their associated intrinsic magnetic properties. Gaining an understanding of these interactions is the key to deciphering the origins of a permanent magnets' magnetic performance and facilitate the engineering of much improved-performing magnets. Here, we use advanced multiscale microscopy and microanalysis on a bulk Sm₂(Co,Fe,Cu,Zr)₁₇ pinning-type high-performance magnet with outstanding thermal and chemical stability. Comparison of the microstructure in regions of different composition, we demonstrate that the pinning of magnetic domains, imaged by nanoscale magnetic induction mapping, is controlled by the composition and atomic arrangement of copper. This is confirmed by micromagnetic simulations. Contrary to the belief that grain boundaries are "weak links" in magnetic materials, we demonstrate grain boundaries undergo magnetization reversal at relatively low fields (0.1-0.3 T), but this remains confined to these regions and does not significantly impact the magnet's coercivity. Our results showcase that it is the optimal microstructure within the grain itself that is crucial for achieving the desired magnetic properties.

Rare-earth-containing permanent magnets for high-efficiency electric motors are critical to the successful electrification of transportation, and improving their operational performance becomes a key challenge of the transition to net-zero carbon emissions. These hard magnets are intentionally engineered with a complex microstructural arrangement of multiple phases and interfaces, each possessing distinct chemistry, structure, unique intrinsic properties and that self-assemble at the nanoscale during the magnet fabrication process[1,2]. The interplay between these individual building blocks is what controls the magnetic domains nucleation and pinning[3], and therefore provides the material with its bulk magnetic performance. Microstructural information at multiple length scales is key to properly understanding

A full list of affiliations appears at the end of the paper. ✉e-mail: leopoldo.molina-luna@aem.tu-darmstadt.de; baptiste.gault1@univ-rouen.fr; konstantin.skokov@tu-darmstadt.de

these microstructural components in bulk magnets that can either degrade performance (weak links) or improve it (perfect/functionalized microstructural components). All possible weak links must hence be identified, particularly to derive measures to mitigate possible negative impact on performance, or better even further improve properties. This often requires correlations of microstructural and magnetic information across multiple length scales to help establish guidelines for science-driven design of permanent magnets, to avoid slow and costly empiricism. However, because of the scales involved, down to the nanoscale, this remains arduous and often challenging.

To date, there are two main classes of high-energy permanent magnets based on the NdFeB- and SmCo-systems. Globally, the latter occupies a relatively small market share compared to the former[4]. Yet the relatively low thermal stability of Nd-Fe-B[5,6] precludes their use in lightweight high-speed electric motors, whereas $Sm(Co,Fe,Cu,Zr)_{7\pm\delta}$ outperforms any other permanent magnet at temperatures above 250 °C[7–10]. Its corrosion resistance makes it near-immune to its chemical operating environments and therefore becomes indispensable for high speed/high power electric vehicles and aeronautic applications. However, Sm and Co are critical elements[11], with growing economic importance and supply risk[12], with severe ethical and environmental issues associated with their mining and extraction.

Sintered $Sm(Co,Fe,Cu,Zr)_{7\pm\delta}$ ($\delta = 0.1...1$) magnets consist of grains of about 100 μm in diameter, usually textured along the c-axis of the matrix $Sm_2Co_{17}$ phase. These grains have a microstructure on the nanoscale, which is indispensable to achieve high coercivity[13–17]. A matrix $Sm_2(Co,Fe)_{17}$ (2:17 R or short 2:17) phase is subdivided into cells of about 100 nm in width by the $Sm(Co,Cu)_5$ (1:5H or short 1:5) cell boundary phase of 10 nm thickness. Both phases are intersected by the Zr-rich (Z) phase forming lamellae thinner than 10 nm, perpendicular to the c-axis of $Sm_2Co_{17}$[18,19]. This self-assembled microstructure consisted of these three ferromagnetic and exchange-coupled phases forms upon complex heat treatment of the magnets and enables high performance[18,20–22]. Sustainable design strategies have involved for instance Co substitution by non-critical Fe, which leads to higher remanence and higher energy densities[23], unfortunately, a content of Fe above 20 wt.% worsens both the optimum squareness of magnetization curves and coercivity[24]. Further substitution of Co by Fe is necessary to increase the remanence, however, it requires adjustment of the composition and processing to maintain this delicate microstructure and, hence, coercivity[25–29].

The production of Sm-Co-Fe-Cu-Zr magnets involves a complex metallurgical process, that includes milling, compaction in magnetic field, sintering, high-temperature homogenization, isothermal aging, slow cooling, and second-step aging. During sintering, compositional heterogeneity develops between grain interiors and grain boundaries as a consequence of solid-state diffusion and grain growth. In particular, the redistribution of transition metal elements is strongly influenced by Fe content: higher Fe concentrations enhance diffusion kinetics and promote preferential segregation at grain boundaries, thereby accentuating the difference in chemical composition between bulk grains and boundary (GB) regions[22,29]. The GB area is Cu-depleted and includes magnetically softer phases (e.g., $Sm_{n+1}Co_{5n-1}$), precipitate-free-zones and other defects and microstructural features[30,31]. Near the GBs of sintered magnet, the 1:5/2:17 cellular nanostructure is coarse and inhomogeneous, and this fact have been recognized as important microstructural origins for the lower-than-ideal energy product of precipitate-hardening Sm-Co-Fe-Cu-Zr magnets with high Fe content[24,28,32,33]. In addition, magnetization reversal in the GB region occurs in relatively weak fields (0.1–0.3 T), causing a distinctive kink in the hysteresis loop, while the rest of the sample demagnetizes at much stronger fields, closer to $H_c$ (2–3 T)[24,28,32,33]. This is why GBs are often recognized as "weak links" in terms of coercivity. However, a detailed study reporting the complete evolution of the demagnetization process, in relation to the characteristic microstructure and composition of defects is missing, leaving a knowledge gap regarding their true influences the final coercivity and energy product and hindering more precise microstructural design of permanent magnets.

Here, we investigated the correlations between microstructure and hard magnetic properties of high-end, production-grade $Sm(Co_{0.65}Fe_{0.27}Cu_{0.06}Zr_{0.02})_{7.7}$ sintered permanent magnets with a high Fe content using Kerr microscopy, magnetic force microscopy (MFM), scanning electron microscopy (SEM), atomic resolution transmission electron microscopy (TEM), atom probe tomography (APT), Lorentz microscopy and electron holography. Two magnets from the same industrial batch were selected for the study: sample A showing the lowest coercivity in the batch ($\mu_0 H_c = 2.2$ T) and sample B (reference magnet) with optimal properties and maximal performance ($\mu_0 H_c = 3$ T). All production parameters were identical, except for the high-temperature homogenization step: magnet A with lower coercivity, was annealed at a temperature 7 K higher than magnet B, produced under optimal conditions. Given that such temperature deviations can occur in large-scale technological processes, it is essential to understand the corresponding microstructural changes that lead to a reduction in coercivity.

As a result of sintering, followed by multi-step annealing treatment, low coercivity regions ($\mu_0 H_c \sim 1$ T) appear mostly near the grain boundaries in sample A. Despite a rather similar geometry of the cellular nanostructure in both regions of sample A, the chemical composition and distribution of elements within the main phases differ significantly for the high-coercivity and low-coercivity regions. Comparing our experimental results with micromagnetic modelling, we can reveal that the characteristic features of the microstructure responsible for the formation of high coercive state in the $Sm(Co_{0.65}Fe_{0.27}Cu_{0.06}Zr_{0.02})_{7.7}$ magnet with high Fe concentration: these are the (widely known) Cu-rich cell boundary 1:5 phase and the (identified in this study) Cu-rich coating layers of the 1:5 phase at the Z-platelets – acting as 'functionalized defects' to improve its magnetic properties. Furthermore, contrary to the common belief that grain boundaries with incomplete cellular structure are the "weak link" in Sm-Zr-Co-Cu-Fe magnets, our findings suggest otherwise. Although the GB region experiences magnetization reversal at relatively low fields (0.1–0.3 T), this reversal is limited and confined to the thin outer layer of the grains, and does not have a significant effect on the magnet's overall coercivity. Nevertheless, even a thin GB region can slightly reduce the squareness factor of the demagnetization curve and thereby lower the maximum energy product. However, since such regions are unavoidable due to the intrinsic limitations of the technological process in magnets A and B, the resulting reduction in properties from the GB phase is relatively minor. Our results highlight that achieving the desired magnetic properties is primarily dependent on the optimal microstructure within the grain itself.

## Results

To understand the complex microstructural features and their influence on the overall magnetic performance of Sm-Co 2:17-type permanent magnets, a multiscale characterization approach is essential. Backscattered electron (BSE) scanning micrograph in Fig. 1a, shows multiple grains of the A magnet, and we focus here on two areas close to grain boundaries inside of the grains, which outlined in red, appear brighter, indicative of a higher average atomic weight, i.e. a higher Sm content. Kerr microscopy of the thermally demagnetized sample (Fig. 1b) reveals fine domains in these specific regions. Magnetization in a pulsed field of 7 T and application of demagnetizing field pulses of −0.5 T (Fig. 1c), and −0.9 T (Fig. 1d), evidence the existence of regions that appear dark, where demagnetization initiates and hence exhibit a relatively lower coercivity compared to the bulk of the grains, that are not demagnetized in these fields. In the following, the dark regions and the bright grain bulk regions are referred to as low $H_c$ and high $H_c$ regions, respectively. The higher Sm content in the low $H_c$ regions is

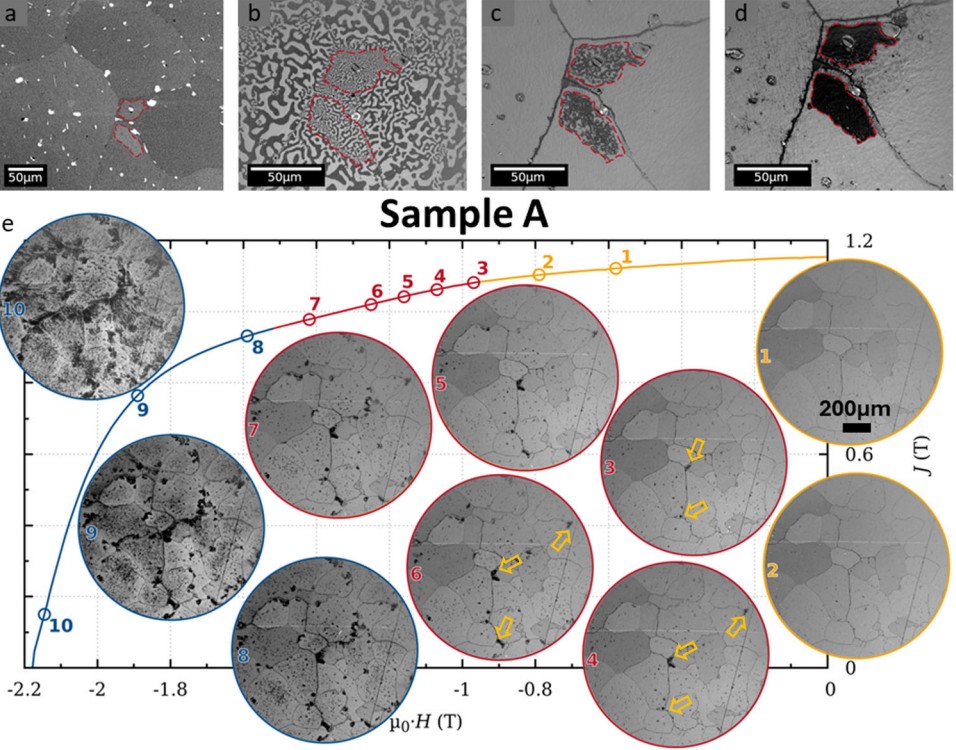

**Fig. 1 | Magnetic domain and microstructure imaging of sample A. a** BSE-SEM image showing the grain structure, including two intragranular regions marked in red in neighbouring grains of magnet A. **b–d** Series of Kerr microscopy images illustrating the change of magnetization between the regions of interest in the demagnetized state (**b**) The demagnetization starts at 0.5 T (**c**). The distorted areas are fully demagnetized at 0.9 T (**d**). **e** Demagnetization of the sample after applying a field of 14 T. The insets show Kerr micrographs taken after subsequent demagnetization. Each image corresponds to the state indicated by a circle on the polarization curve J(T). Nominal c-axis is out of plane for all images. Arrows in some of the images indicate regions where demagnetization initiates and first propagates.

confirmed by energy-dispersive X-ray spectroscopy (EDS), as reported in Tables S1–S3. Additional measurements an images are reported in Figures S1, S2a–f, S3, S4.

Figure 1e shows the second quadrant of the hysteresis loop of magnet A measured at 300 K, and the inset shows Kerr micrographs after application of demagnetizing fields along direction of the texture (c-axis out of plane). GBs appear as weak points where magnetization reversal starts[20,34], but no intragranular magnetization reversal appears up to fields of 0.9 T (yellow curve, (1)-(3)). For 1.0 T, the magnetization of several low $H_c$ (dark) regions starts to be partially reversed inside grains and near GBs. With higher demagnetizing fields the magnetization reversal spreads inside the affected areas and occurs in more of these low $H_c$ regions (red curve (3)-(7)). Within an investigated area of 2.4 mm², the first magnetization reversal outside of a low-$H_c$ region appears for a demagnetizing field of 1.5 T (blue curve (8)-(10)), which correlates with the onset of a drastic reduction in magnetic polarization J of the whole sample.

A similar approach was used for magnet B, microstructural and magnetic domain analyses of the reference sample are summarised in Fig. 2. As can be seen from the BSE-SEM image (Fig. 2a), a homogeneous microstructure is observed with no visible change in contrast relative to different chemical composition. The interaction-type magnetic domain structure, imaged by Kerr microscopy, in the thermally demagnetized state (Fig. 2b) appears uniform across the sample, with no significant change in domain width size. After saturating the sample in a 7 T magnetic field, the sample was subjected to demagnetizing field pulses of − 0.5 T (Fig. 2c), and − 0.9 T (Fig. 2d). One can observe the dark contrast only at the grain boundary region, indicating a magnetization reversal confined along these areas with no significant difference at higher applied magnetic field, as expected from its high

coercivity value. In contrast to magnet A, no additional magnetically reversed areas are observed, confirming the uniformity of the microstructure.

Figure 2e shows the room temperature demagnetization curve of the reference sample B and, in the inset, Kerr micrographs after application of demagnetizing fields along the easy magnetization direction starting from saturated state. As previously noted, the sequence of Kerr micrographs further confirms that the magnetization reversal starts at the grain boundary region. When a sufficiently high demagnetization field is applied, the domain wall pinning energy is exceeded and the reversal magnetic domains grow in a stripe-like fashion, resembling beach sand ripple marks. This arrangement was only observed when the demagnetizing field is around to −2.5 T (inset 4 in Fig. 2e), which is close to the kink observed in the hysteresis loop (point 4). It is worth noting that, despite the strong texture along the easy magnetization direction (out of plane of the image), the magnetic domain stripes within each grain exhibit distinct orientations.

The clear identification of regions of low coercivity only found in magnet A is a unique opportunity for a direct comparative study to reveal the microstructural parameters underpinning differences in properties. For transmission electron microscopy and atom probe tomography, samples were extracted from areas of both types that had previously been identified by SEM and Kerr microscopy. Low-magnification bright-field (BF) TEM images show a well-defined cellular nanostructure in both regions of magnet A, as seen in Fig. 3a, b, respectively, which comprise a network of 1:5 cell walls surrounded by pyramidal 2:17 cells, intersected by the Z-platelet phase. Cells are 272 nm wide on average for the high-$H_c$ region compared to 238 nm for the low-$H_c$ region (Table S1). The number of Z-platelets per area is considerably higher in the low-$H_c$ area, while both 1:5 and Z phases are

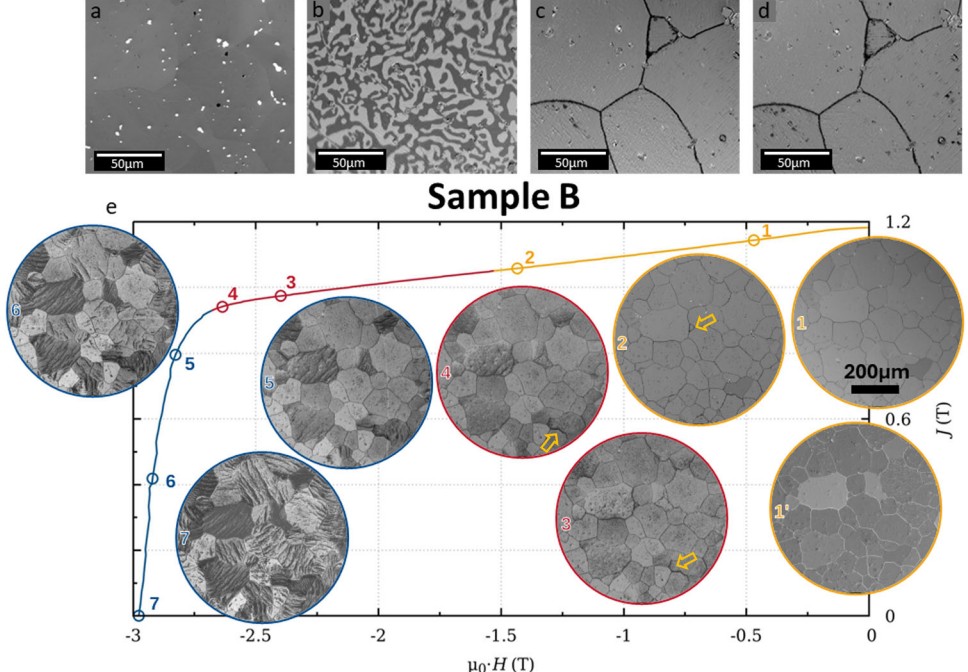

**Fig. 2 | Magnetic domain and microstructure imaging of sample B. a** BSE-SEM image showing homogeneous grain structure of the reference magnet B. **b–d** Series of Kerr microscopy images illustrating the change of magnetization between the regions of interest along the easy magnetization direction. **b** Thermally demagnetized state and after applying a demagnetization field of 0.5 T (**c**) and 0.9 T (**d**) from a saturated state. **e** Demagnetization of the sample after applying a field of 7 T. The inset images show Kerr micrographs taken after subsequent demagnetization. Each image corresponds to the state indicated by a circle on the polarization curve J(T) – additionally one Kerr micrograph was taken after applying 7 T in the reverse direction, as can be denoted for the contrast change between inset 1 (dark contrast at the grain boundary) and 1' (bright contrast at the grain boundary). Nominal c-axis is out of plane for all images. Arrows in some of the images indicate regions where demagnetization initiates and first propagates.

thicker in the high-$H_c$ area. The selected-area electron diffraction (SAED) patterns for both regions (Fig. 3a1, b1) are rather similar. A detailed analysis (Fig. 3a2, b2, d) reveals that a small volume fraction of the intermediate 2:17 R' phase is left from an incomplete transition of the high temperature 2:17H to the low temperature 2:17 R phase. 2:17 R' was shown to have a negative influence on the coercivity[35–37]. In addition, analysis of the streaks in Fig. 3a3, b3 confirm the higher density of the 1:5 phase in the low $H_c$ area, while Fig. 3c, the normalized intensity profiles extracted from (000) to (003) reflections of the SAED pattern confirm the higher density of the Z-phase in the low-$H_c$ region. The bimodal distribution points to irregularities in the spacing of the platelets. The profile also shows higher intensity of reflections associated to the 2:17 R' phase at 1/3 and 2/3 in the low $H_c$ area, supporting a higher volume fraction of this detrimental phase.

Figures 4a–c present BF-TEM image, SAED pattern, and high-resolution (HR) TEM image of the reference sample B, respectively. A comparison between the grain size of the reference sample B and the sample A shows that the grains in the former are slightly larger. However, the SAED of pattern of the sample B does not show any specific difference from the pattern of sample A. The same is also observed in the HRTEM image of reference sample B. It shows that 2:17, 1:5, and Z-platelets are well developed, although the density of both phases is slightly lower than that in sample A. Still, no considerable differences are observed between sample A and the reference sample B.

To complement TEM, compositional mapping in 3D at the nanoscale was performed by APT to reveal the change in chemical composition at the nanoscale in sample A. Fig. 5a1, b1 show representative APT reconstructions for the low- and high-$H_c$ coercivity regions of the A magnet. The limited solubility of Cu in the 2:17 and Z-platelet phase and of Zr in the 1:5 and 2:17 phases[38,39], allows us to use

these elements to segment the data: isoconcentration surfaces encompassing regions containing more than 12 at.% Cu in blue and 6 at.% Zr in green, reveal the Cu-rich 1:5 and the Z-phase respectively. Table S3 summarises the compositions and phase fractions derived from the APT data, which are comparable to previous reports[40]. For the Fe-rich 2:17 phase, the cell sizes are obtained by TEM, because the limited volume in a single APT dataset precludes us from estimating their actual size. APT confirms the observations by SEM in Fig. 1: the low-$H_c$ region is richer in Sm and Cu by approximately 1 at.% and poorer in Fe and Co by 1–2 at.% compared to the high-$H_c$ region. Supporting measurements by SEM-EDS can be found in Table S3.

In the reconstructed APT data, the approx. 12 nm-thick 1:5 phase forms a 3D network interrupted by the Z-phase platelets with a thickness of near 10 nm. Compared to the high-$H_c$ region, in the low-$H_c$ region, the 1:5 phase is closer to an ideal diamond shaped structure[19], with a relatively denser network, with 8% more of the 1:5 phase and 5% more of the Z-phase (Table S3). The composition of the 2:17 phase is compatible within the two areas, see Table S3. In contrast, across representative 1:5-phase plates (dashed blue boxes in Fig. 5a1, b1), profiles (yellow arrows in Fig. 5a2, b2) reveal a composition lower by 5 at.% in Cu and higher by 4 at% in Co in the low-$Hc$ region. Composition profiles through representative Z-platelets (dashed green boxes in Fig. 3a1, b1), plotted in Fig. 5a4, b4 reveal two striking differences between the high-$H_c$ and low-$H_c$ areas: the Cu concentration increases in the centre of the Z-platelet by up to 4 at.% and the Z-2:17 phase boundary is enriched up to 8 at.% Cu in a ~5 nm thin region for the high-$H_c$ area. A two-dimensional view in Fig. 5b5 suggests that this Cu-enrichment stems from a Cu-rich coating layer which is discontinuous. A similar increase in Cu at the Z-platelet–2:17 boundary is reported in ref. 41, and a lower but detectable increase in Cu at the Z-platelet interface was also observed in ref. 42, but neither are discussed any

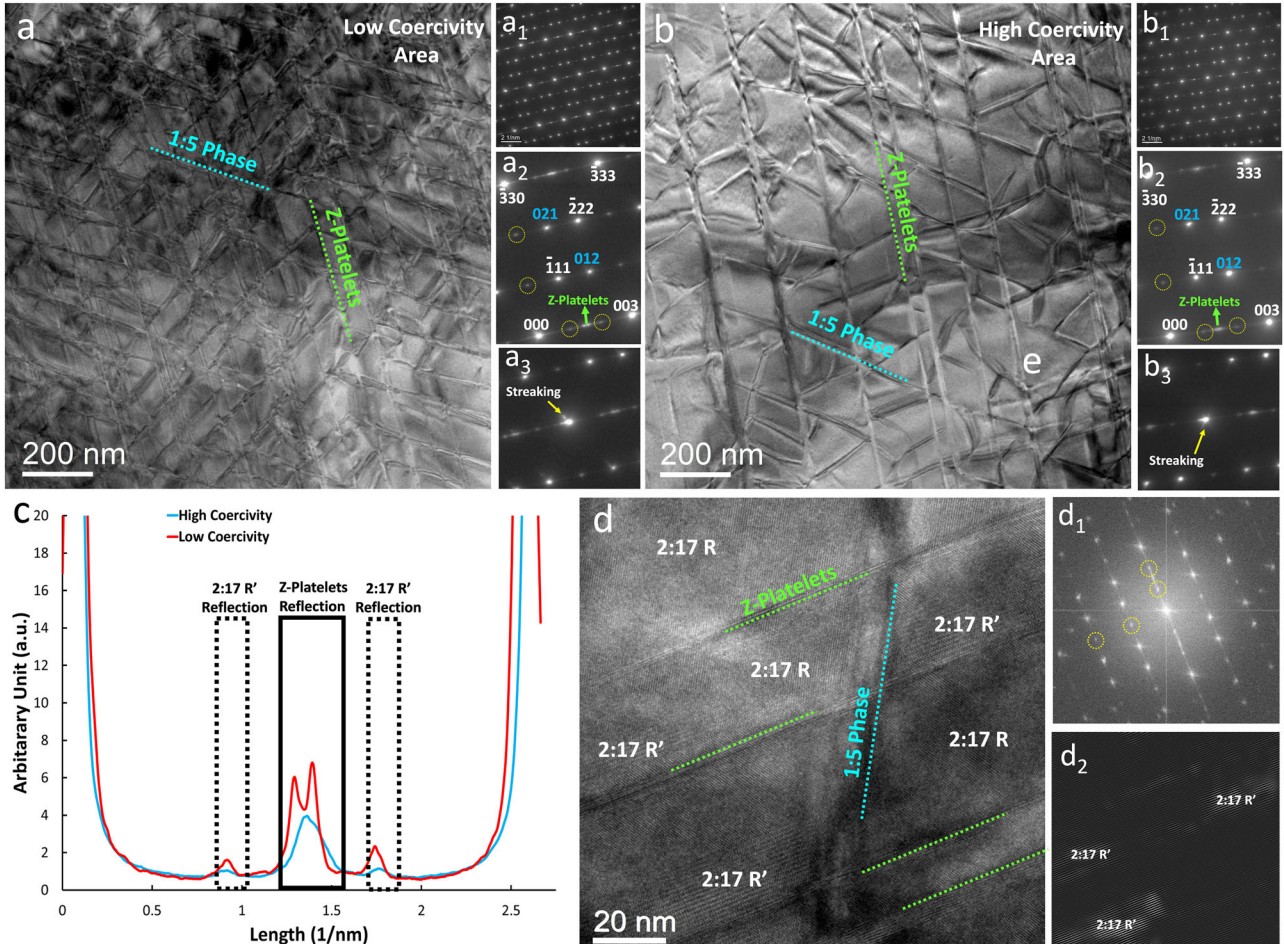

**Fig. 3 | Nanoscale phase distribution in high and low coercivity regions of sample A.** BF-TEM images of magnet A (**a**) low- and (**b**) high-$H_c$ regions with corresponding electron diffraction patters (**a1–a3**) and (**b1–b3**), respectively) taken in [110] zone axis of the 2:17 phase. White colour indices are used for 2:17 R and blue colour indices for 2:17 R twins, respectively. Yellow circled reflections belong to the 2:17 R' phase and Z-platelets are indicated with green arrows. The normalized intensity profile along c-direction (**c**) outlines the higher amounts of 2:17 R' phase and Z-platelet phase in the low-$H_c$ region. **d** HRTEM image from the high coercivity area showing the presence of 2:17 R' phase in the nanostructure. (**d1**) is FFT image of (**d**), and (**d2**) is inverse FFT image of (**c**), filtering only reflections of 2:17 R' phase.

further. Overall, the high-$H_c$ region of the magnet A contains less of a 1:5-phase that is richer in Cu, and has fewer Z-platelets, but also with increased Cu concentrations, both inside the platelets and at the interfaces to the 2:17 phase.

In sample B, APT analyses were performed in the center of the grain and at a GB, with representative reconstructions shown in Figs. 6, 7, respectively. The grain center exhibits a fully developed nanoscale structure with Fe-rich 2:17 phase cells, Cu-rich 1:5 phase cell walls, and Zr-rich Z-platelets, Fig. 6a, highly similar to the high $H_c$ areas of the A sample: (1) The ~10 nm thick 1:5 phase shows a high Cu concentration peaking at ~30 at.%, as shown by 2D-Cu profile and 1D profile in Fig. 6b, c; (2) Z-platelets have discontinuous Cu-rich coating layers peaking at 8 at.% Cu (Fig. 6d), with thicker platelets ~10 nm exhibiting Cu-enriched centers (Fig. 6e), that are not found in the thinner platelets ~ 5 nm (Figure S6 in the supplement).

The GB is primarily the Fe-rich 2:17 phase without a well defined cellular or lamellar structure (Fig. 7). At a low isosurface value of 5 at.% Cu, features resembling residual cellular and lamellar structure and a Sm, Zr, and Cu-enriched particle become apparent as seen in Fig. 7a. Two parallel planar and discontinuous features, resembling the Z-phase, exhibit a Cu-rich coating layers, compatible to Z-platelets in the grain center, along with Zr and Co composition variations, as seen in a representative 1D-profile of one "Z-platelet" (Fig. 7b). Similar

features were observed in the GB of the sample A by scanning-TEM (STEM)-EDS (Figures S7, S8). Another set of 8 nm thick planar features forms a junction, resembling the 1:5 phase with low Cu-content peaking at 8 at.%, as a representative 1D profile in Fig. 5c shows. Additionally, a ~ 100 nm large particle enriched in Sm, Zr, and Cu with a composition $Sm_{16}Co_{56}Fe_{12}Zr_7Cu_8$ was partially captured (Fig. 7d), inside which appears a set of Zr-rich and Cu-rich layers around 5-10 nm thick, consistent with STEM-EDS results in the GB of the sample A (Figure S8).

TEM and APT observations reveal that the nanoscale structure of the grain interior in magnet B closely resembles that of the high $H_c$ region in magnet A. The main distinction between the samples A and B lies in the presence of low $H_c$ areas in magnet A, consistent with SEM and Kerr results (Figs. 1, 2). Magnet B thus can be considered consisting of only grains with high $H_c$ nanoscale structure.

Figure 8a–b are representative atomic resolution high-angle annular dark-field scanning-TEM (HAADF-STEM) images from the low- and high-$H_c$ regions of the magnet A, respectively, showing a considerably thicker Z-platelet in the latter. The phases are identified based on the atomic stacking (Figure S5) and interfaces are marked with dashed yellow lines. A few layers of 1:5 phase are observed between two Z-platelets (Fig. 8a, c) in low-$H_c$ region. In the high-$H_c$ region, a distinct layer is observed at the 2:17–Z-platelet phase

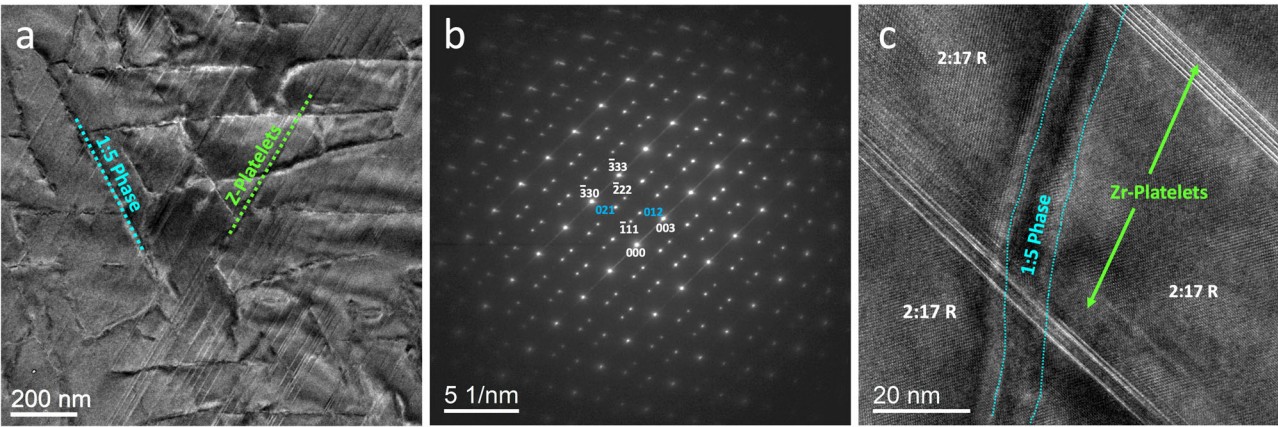

**Fig. 4 | nanoscale phase distribution of sample B. a** Low-magnification bright-field TEM image, (**b**) electron diffraction pattern of the reference sample, and (**c**) HRTEM image of the sample B, showing 2:17, 1:5, and Z-platelets phases.

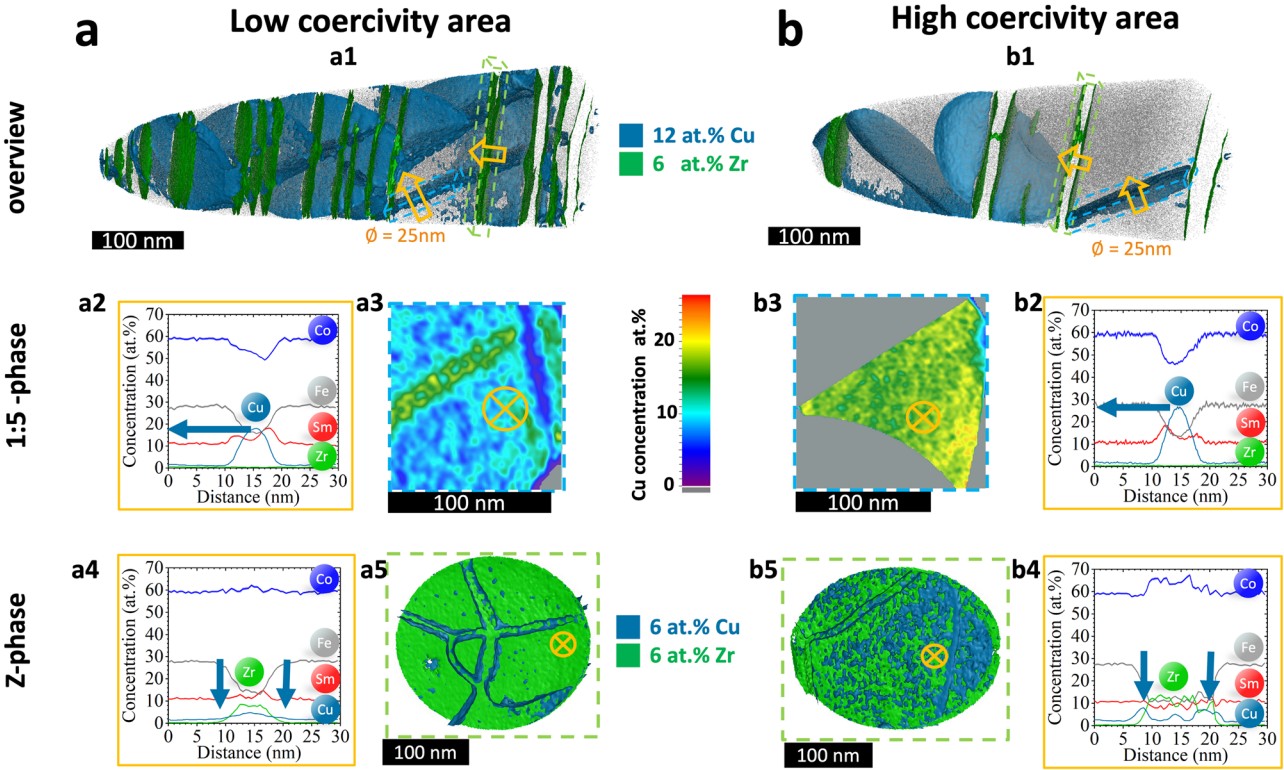

**Fig. 5 | intragranular nanoscale compositional mapping in high and low coercivity regions of sample A.** 3D reconstruction of APT data of sample As for (**a1**) low and (**b1**) high $H_c$ areas showing the geometrical distribution of the 1:5 phase (blue, isoconcentration value 12 at.% Cu) and the Z-phase (green, isoconcentration value 6 at.% Zr). Blue and green dashed lined boxes indicate a representative Cu-2D distribution for the 1:5 phase-plates (**a3** and **b3**) and a rotated 3D reconstruction of the Z-phase-plates (**a5** and **b5**), respectively. The 1D composition profiles are measured perpendicular to the representative 1:5 phase-plates (**a2** and **b2**) and the Z-platelets (**a4** and **b4**) as indicated by the orange arrows in the top row and crossed circles in the nearby figures in the middle and bottom rows. APT data from the high coercivity area shows increased Cu-concentration in the 1:5-phase and at the phase boundary of Z-phase and 2:17 phase (indicated by blue arrows in middle and bottom rows).

boundary (Fig. 8b, d), and the intensity profile in Fig. 8e suggests that is a superposition of the 2:17 and 1:5 phase. These observations agree with a discontinuous Cu-rich coating layer of the 1:5 phase seen by APT (Fig. 5b5).

The link between the differences in microstructure across regions and the magnetic response remains, however, an open question. We address this by imaging the domain walls using magnetic imaging in the TEM at the magnetic remanence and correlate them to the microstructure. Figure 8f–h show a pair of magnetic domain walls formed in the low-$H_c$ region of sample A. The pair of Fresnel images in

Fig. 8f confirm the characteristic zig-zag shape of the walls due to the pinning at the phase boundaries. The direct relationship between the magnetic domain walls and the organisation of the 1:5 and Z-phases phase distribution was imaged using off-axis electron holography, as shown in Fig. 8g, h. The location of the 1:5 and Z-phases can be identified in the in-focus amplitude image (Fig. 8g) and the 1:5 together with the domain wall location marked in the in-plane magnetic induction map (Fig. 8h), which is generated using the magnetic phase shift image extracted from off-axis electron holography measurements. The high-angle magnetic domain walls are pinned at the boundaries of the 1:5

# Grain center

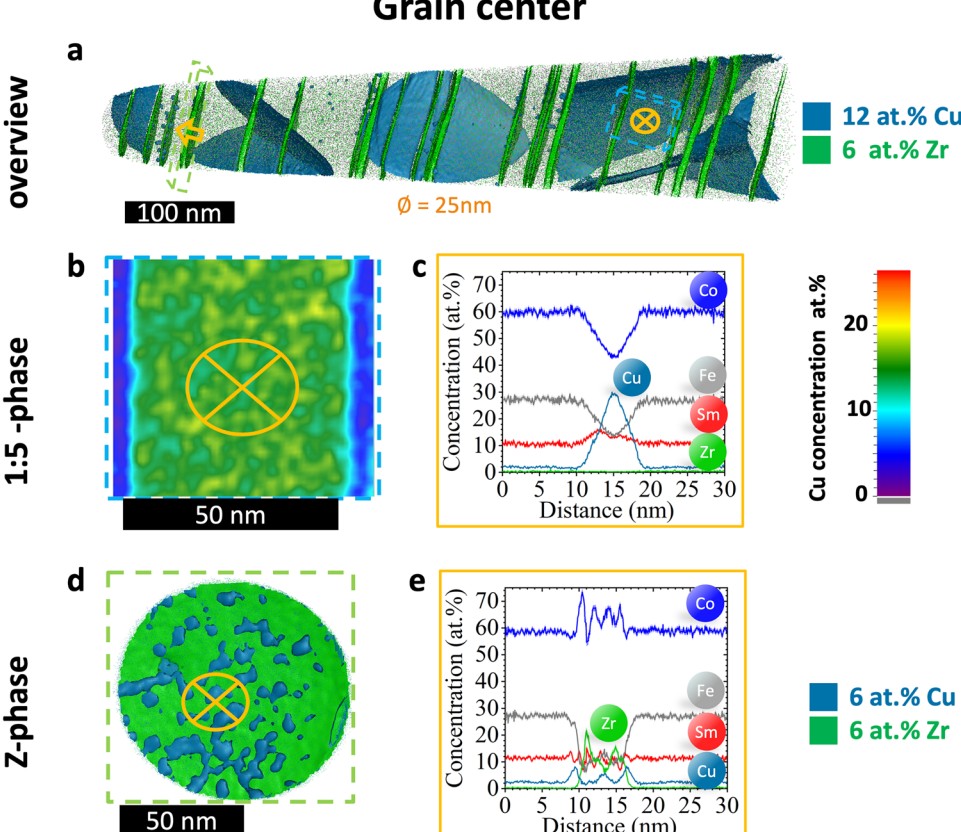

**Fig. 6 | intragranular nanoscale compositional mapping in sample B.** APT data for the grain centre of the reference sample B. **a** 3D reconstruction showing the geometrical distribution of the 1:5 phase (blue, isoconcentration value 12 at.% Cu) and the Z-phase (green, isoconcentration value 6 at.% Zr). Blue and green dashed lined boxes indicate a representative Cu-2D distribution for the 1:5 phase-plates (**b**) and a rotated 3D reconstruction of the Z-platelets (**d**), respectively. 1D composition profiles are measured perpendicular to the representative 1:5 phase-plates (**c**) and the Z-platelets (**e**) as indicated by the orange arrows and crossed circles in (**a**, **b**, **d**). The nanoscale cellular lamellar microstructure in the grain centre of the sample B highly resembles that in the high $H_c$ area in the A sample.

and 2:17 phases. The Z-platelets and 1:5 phase form a complex pattern, which results in an equally complex domain wall pinning and zig-zag shape. These results reinforce the need to control composition, dimensions and distributions of the 1:5 phase and Z-platelets in the 2:17 matrix. In addition, the magnetic states of the low- and high-$H_c$ regions of the A magnet including the grain boundary area were further studied in a single TEM lamella (Figures S7, S8). The structure and chemical composition measurements reveal the lack of the cellular structure in the grain boundary region, Fe-poor composition and in some grains, Sm, Zr and Cu enrichment. These observations together with the measured and relatively large domain wall width in the grain boundary region (Figure S7) suggest a softer magnetic behaviour than that in the low and high $H_c$ regions of magnet A and grain interior of magnet B, which matches well with the Kerr microscopy experiments (Figs. 1, 2).

Our detailed investigation was used to parameterise micromagnetic simulations to link the observed differences in microstructure and magnetic properties of the high- and low-$H_c$ regions in magnet A and the GBs across multiple scales. Details of the simulation setup can be found in **Section 5** of the Supplementary Material (Figures S9, 10 and Table S5). It is worth noting the Z-platelet, presented in the nanostructure from high-$H_c$ region, is modelled with thin 1:5 layers coated (hereinafter mentioned as coated Z-platelet). The three types of pinning sites defined in refs. 19,26, namely 1:5–2:17 intersections cell-corners ($P_1$), Z–1:5 ($P_2$) and Z–2:17 intersections ($P_3$) must be complemented by the coated Z-1:5 intersections ($P_2'$) and coated Z-2:17 interfaces ($P_3'$). Figure 9 present the transient domain structure during the migration of the domain wall in the nanostructures from high- and

low-$H_c$ regions in relation to the simulated magnetisation curve in Fig. 9a. With increasing demagnetizing field, the domain wall propagates from the initial nucleated domain and reaches the pinning sites (Fig. 9$b_1$, $c_1$). At a relatively low applied field (−0.50 T), the effective pinning sites in the high-$H_c$ area are mostly $P_1$ and $P_2'$, and in the low-$H_c$ region are the $P_1$ and $P_2$, respectively (Fig. 9$b_2$, $c_2$). With further increase of the external field, the $P_1$ and most of the $P_2'$ sites remain effective up to −1.6 T in the high-$H_c$ region, along with newly created $P_2'$ sites (Fig. 9$b_3$). The latter are still effective at −1.8 T (Fig. 9$b_4$). For the low-$H_c$ region, however, many $P_2$ sites already fail to pin the domain wall from −0.65 T (Fig. 9$c_3$), where the central area is already fully reversed. The remaining $P_2$ sites fail up until −0.95 T (Fig. 9$c_4$). This comparison of the transient domain structures illustrates the significant effect of the $P_2'$ sites in the high-$H_c$ nanostructure on pinning strength. However, the interfaces between Z-platelets and 2:17 phase show no differences in our simulations and neither $P_3$ nor $P_3'$ pinning sites seem to be relevant for the domain reversal. Additional simulations, in which microstructural parameters such as Z-platelet thickness, cell size, Cu concentration in the 1:5 phase, and the presence of coating layers are systematically and individually varied, show that the latter two factors have the greatest impact on the nanostructure's coercivity, while the effects of the first two are negligible (Figure S11).

To study the demagnetization behaviour in a vicinity of a grain boundary, we performed a simulation on a nanostructure, unveiling a transition between the grain interior with complete 2:17 cells and coated Z-platelet and the GB, approximated by incomplete 2:17 cells and uncoated Z-platelets. The magnetization curve of the nanostructure from the transition region is illustrated in Fig. 9a together with

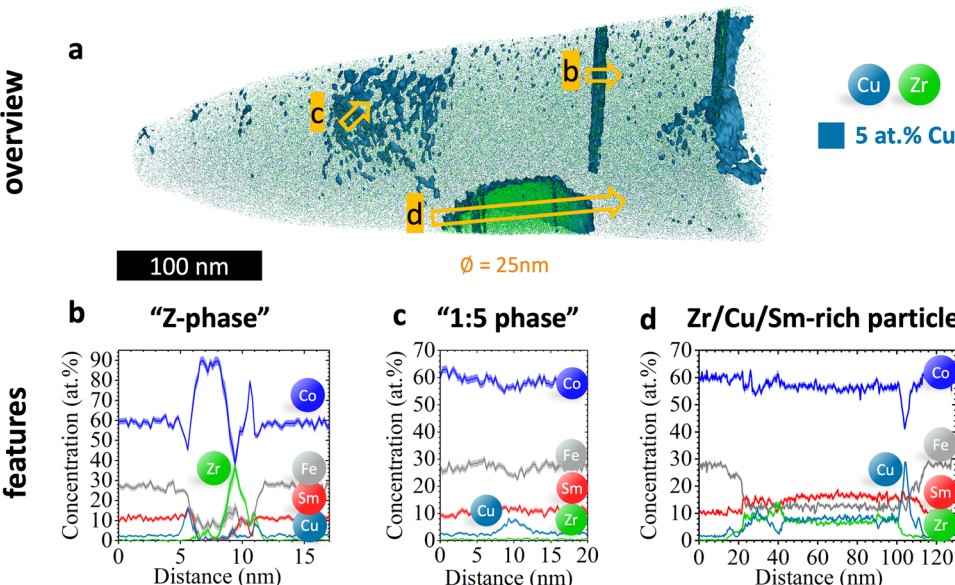

**Fig. 7 | grain boundary nanoscale compositional mapping in sample B.** APT data for the grain boundary (GB) of the reference sample B. **a** 3D reconstruction with Cu-rich isosurface in blue (isoconcentration value 5 at.% Cu) and blue/green points representing Cu/Zr atoms, showing features of incomplete cellular ("1:5 phase", #1) and lamellar structure ("Z-phase", #2) and a Sm, Zr, and Cu-enriched particle (#3) with corresponding 1D profiles (**b**–**d**) across the respective features as indicated by named orange arrows.

those from high- and low-$H_c$ regions, where several typical pinning events during the domain wall migration are presented in Fig. 9d$_1$–d$_4$. For a nanostructure with incomplete 2:17 cells, there are no P$_1$ type and relatively fewer P$_2$ type pinning sites, thus it gets more magnetized (or domain wall propagates comparably further) than these with complete 2:17 cells at the same external field −0.5 T (Fig. 9d$_2$, c$_2$). Meanwhile, as no coated Z-platelet was present in the GB nanostructure as well, the domain wall pinned by P2 sites is swiftly de-pinned and migrated into the grain interior with the complete nanostructure, where it is pinned by P$_1$ and more effective P$_2$′ sites (Fig. 9d$_3$). As a consequence, a reduced migration speed of domain wall from nanostructures with incomplete- to complete 2:17 cells is depicted by the magnetization curve (Fig. 9a). This phenomenon reveals easy demagnetization in the GB region, which is significantly slowed down in the grain interior, highlighting the significant influence of the nanostructural phase formation on the domain wall migration, and further on the coercivity.

In conclusion, we set out to unveil the mechanisms underpinning the performance of bulk Sm$_2$(Co,Fe,Cu,Zr)$_{17}$ magnets, i.e., a quest to identify the weaker link and the functionalized defect in the complex assembly of building blocks that combine to provide a magnet with its set of physical properties of interest. Our comparative multiscale analyses of the distribution of the different phases, their structure and composition demonstrate how small differences in the local composition and atomic arrangement of the phases can significantly alter the microstructure and hence the overall performance. Schematics of the microstructures based on our findings can be found in Figure S12a–b for the low and high coercivity regions, respectively. An unexpected combination of the Z-phase with the 1:5 phase forming a Cu-rich coating layer is only found in the high-$H_c$ region of magnet A and throughout grain interior of the magnet B (Figure S12b).On the one hand, the beneficial character of such a microstructure may be understood by the large difference in domain wall energy of these two phases, forming an optimal pinning centre[19] stretched out over the whole area of the Z-platelets instead of being limited to the intersections of the 1:5 phase and the Z-platelet phase, as previously reported[26].

On the other hand, a higher Cu concentration inside the 1:5 phase in the high $H_c$ regions additionally enhances the pinning effect[21,42–44]. The complex role of Cu has already been discussed previously, yet there remain many opened questions[45–47].

Here, the combination of both, Cu-rich 1:5 cell boundaries and Z-phase with Cu-rich 1:5 phase coating layers and its different intrinsic properties, explains the higher coercivity in the corresponding regions, thus both qualifying as "functionalized defects". Interestingly, this holds true even though both phases are present with a lower volume fraction and deviate from the ideal geometry. This indicates the importance of the composition over geometrical differences of the 1:5 cell boundary and Z-platelet phase not only for individual pinning sites but the overall pinning effect and thus the resulting coercivity. The identification of the 1:5 phase coating layers *on the surface* of Z-platelets as active pinning centers provides an engineering approach to enhance coercivity in 2:17-based SmCo magnets. This approach involves optimizing the pinning properties of the 1:5 coating layers by refining their geometry and composition to achieve values comparable to the cell boundary 1:5 phase. Specific improvements include ensuring the 1:5 coating forms a continuous layer (currently discontinuous), increasing the Cu concentration to >20 at.% (currently ~8 at.%), and enhancing the thickness to 10 nm (currently 5 nm) as schematically shown in Figure S12c. Achieving these improvements will likely require adjustments to alloy composition and heat treatment procedures. If the 1:5 phase interlayers *within* the Z-platelets also act as additional pinning centers—a hypothesis that requires further investigation—similar engineering strategies could be applied to these interlayers to further enhance coercivity in 2:17-type magnets. The presence of low-$H_c$ regions with suboptimal nanoscale structure and microchemistry can hence be considered a "weak link" of the magnet (Figure S12a). Since the high-coercivity regions in magnets A and B have almost the same composition, it is the presence of low-coercivity regions that is the cause of the overall decrease in the coercivity of the entire sample A, which is also fully confirmed by the results of the micromagnetic analysis.

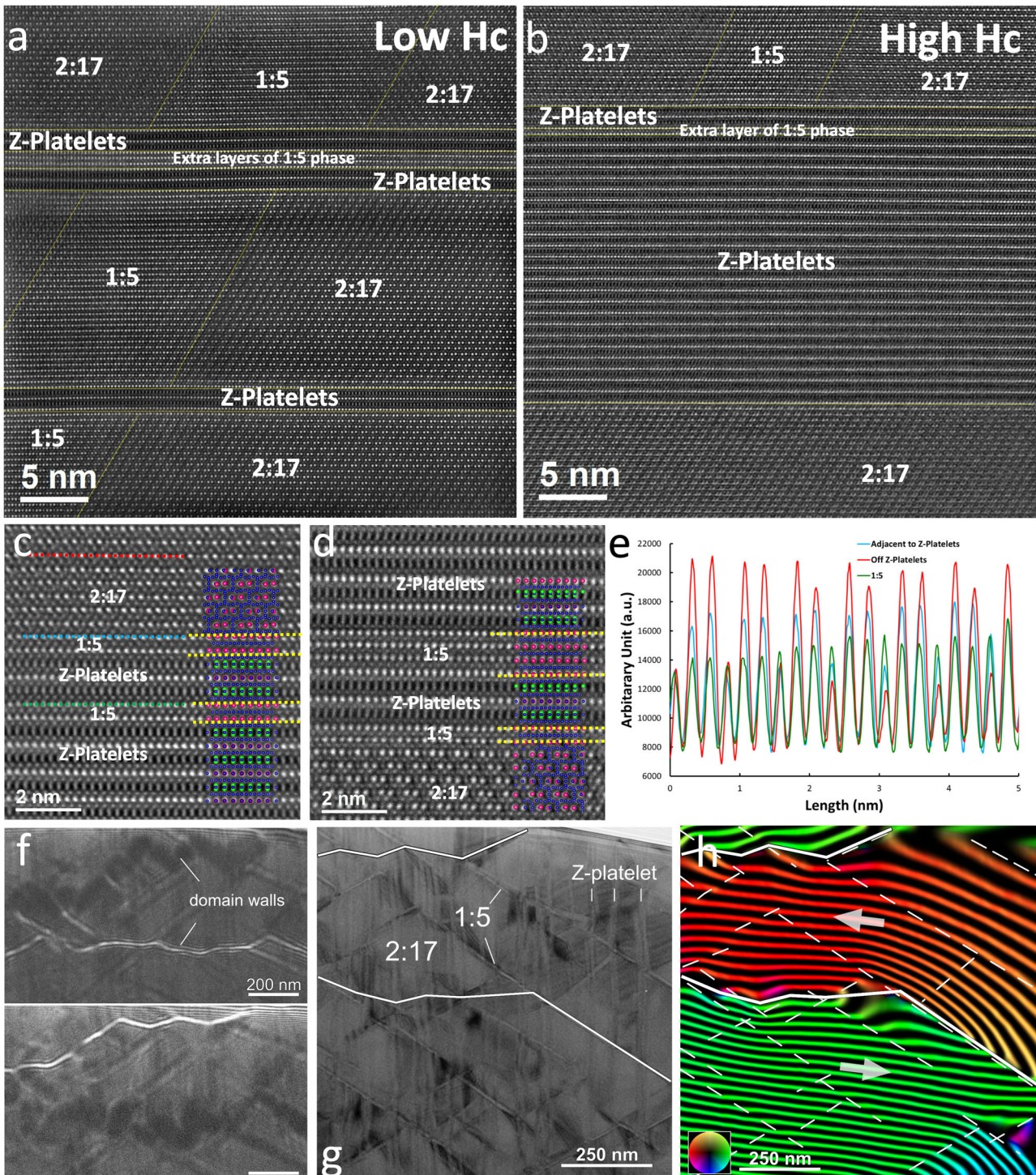

**Fig. 8 | High-resolution structural and magnetic imaging high and low coercivity regions in sample A.** HAADF STEM images of (**a**) low- and (**b**) high-H_c areas of the A magnet. Areas are labelled according to their atomic stacking being matched with theoretical structures (cp. Fig. S5). **c, d** Magnified images highlighting the existence of 1:5 phase as interlayers and coating layers at Z-platelets. **e** Image intensity profiles extracted from dashed lines with corresponding colour in (**d**) show the mixed nature of the coating layer. **f** Under- and overfocused Fresnel images of magnetic domain walls in low-H_c grain. The defocus applied was 0.5 μm. **g, h** An amplitude image and the corresponding magnetic induction map showing the pinning of domain walls at the phase boundaries of the 1:5 and 2:17 phases. Solid and dashed lines mark the domain wall and 1:5 phase locations, respectively. The colours and contours lines indicate the field line direction and strength. The contour spacing is 2π radian.

In addition, contrary to the common belief that the grain boundaries are the "weakest link" in bulk magnets, we have shown that this is not the case. Indeed, the GB region undergoes magnetization reversal in weak fields, (0.1-0.3 T) but the process stops at the boundary of the grain interior, where domain walls meet the nanostructure of the grain and get pinned. This magnetization reversal at GBs does not affect the overall coercivity of the magnet, manifesting itself as a small kink in low negative field in the hysteresis loop (see Figure S1). Our work shows that it is the optimal microstructure inside the grain that is the key to optimal magnet properties. Our study

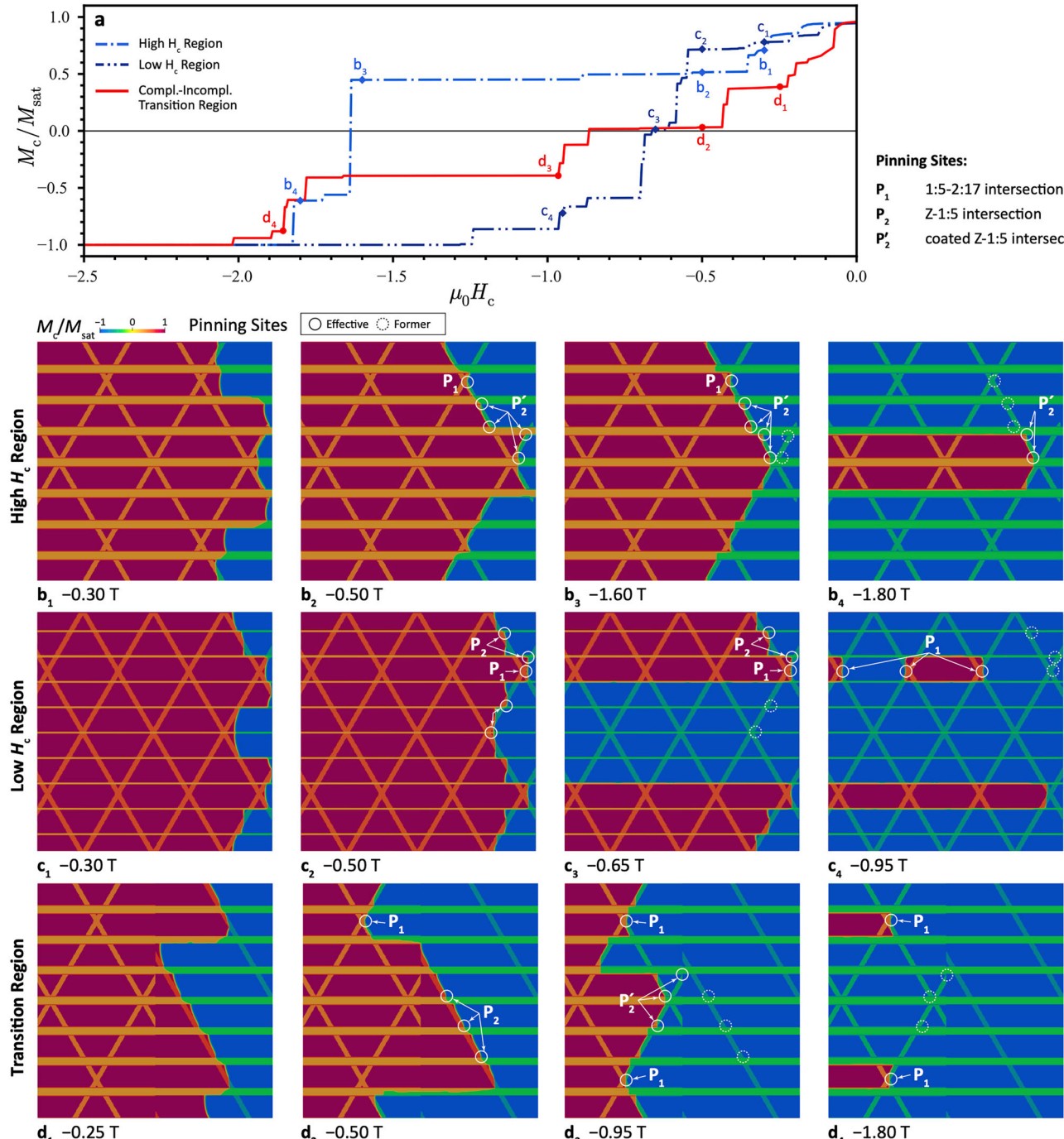

**Fig. 9 | micromagnetic simulations to link microstructure to properties.**
**a** Demagnetization curves of the nanostructures from high- and low-$H_c$ regions as well as transition region using nucleated initial conditions. Transient domain structures with marked pinning sites at selected applied fields $\mu_0H$ marked in (**a**) for (**b**) the high-$H_c$ nanostructure, (**c**) the low-$H_c$ nanostructure, and (**d**) the transition zone between complete-incomplete nanostructure.

highlights the importance of complete identification of each building block constituting these permanent magnets at multiple length scales, and understanding the subtle differences in their arrangements, composition and physical properties. All of this is indispensable for the further improvement of unique characteristics such highly sought-after permanent magnets.

## Methods

Powders of $Sm_2(Co,Fe,Cu,Zr)_{17}$ prepared from book mold ingots by crushing and milling have been ground to an average particle size of about $7 \pm 2\,\mu m$ using a AFG jet mill and blended to obtain the desired chemical composition of about $Sm(Co_{65}Fe_{27}Cu_6Zr_2)_{7.7}$. Green bodies were prepared by alignment of the powder particles in a magnetic field of 1300 kA/m and subsequent isostatic pressing with a pressure of about 250 MPa. The compacted green bodies have been sintered at about 1200 °C and kept slightly below sintering temperature for homogenization. The only difference between the two studied magnets was the high-temperature homogenization step: magnet A, which exhibited lower coercivity, was annealed at a temperature 7 K higher than magnet B, processed under optimal conditions. Subsequent quenching and annealing at 850 °C followed by slow cooling to 400 °C at 0.7 K/min, holding for several hours and final quenching completed

the heat treatment of the samples. Pieces from the center of the sinter body have been prepared for microscopic and magnetic measurements. Three magnets from the same batch with similar composition and processing has been investigated, named A (worst performance), B (reference, best performance) and C (medium performance).

Isothermal magnetization measurements along the c-axis of textured samples were carried out using a commercial vibrating-sample magnetometer (Quantum Design PPMS-14) in steady magnetic fields up to 14 T at ambient temperature (300 K). Magnetic pulses up to 7 T were applied within a commercial pulse field magnetometer (Metis HyMPulse) at ambient temperature.

Scanning electron microscopy (SEM) images were obtained using back scattered electrons in a Tescan VEGA3 SBH and a JEOL JSM-7600F for high resolution images, respectively. Energy dispersive X-ray spectroscopy (EDS) was used in the Tescan microscope with an EDAX Octane Plus detector to obtain overall compositions.

An evico magnetics optical Kerr microscope was used for imaging magnetic domains via the magneto-optical Kerr effect at ambient temperatures. The polar sensitivity was used, so the out-of-plane component of the magnetization is visible in the images. Magnetic force microscopy (MFM) was also employed to examine the region between the high- and low-coercivity areas of magnet B. Measurements were performed using a standard dual-pass mode on a Bruker Dimension Icon device, equipped with a high-coercivity cantilever (ASYMFMHC-R2, Asylum Research, USA).

Electron transparent specimens for TEM were fabricated by Ga focused ion beam (FIB) and plasma sputtering using dual beam SEM/FIB systems (Zeiss Crossbeam 540 and ThermoFisher Helios G4 plasma FIB). Bright-field (BF) TEM imaging and selected-area electron diffraction (SAED) measurements were carried in a conventional transmission electron microscope (JEOL JEM 2100 F). High-resolution high angle annular dark field (HAADF) scanning TEM (STEM) imaging was carried out in an aberration-corrected system (JEOL JEM-ARM200F) operated at 200 kV.

A combined SEM/FIB Dual-Beam Helios Nanolab 600i (FEI) was used to cut needle shaped specimens according to the typically used protocol reported by Thompson et al.[48], from selected regions of the thermally demagnetized and polished magnet using a low energy (5 keV) Ga beam for final milling to minimize beam induced damage. These needles were investigated with a CAMECA LEAP 5000 XS local electrode atom probe at a constant temperature of 60 K under ultrahigh vacuum conditions ($10^{-10}$ mbar) using a pulsed UV laser (355 nm wavelength, 10 ps pulse duration with 45 pJ pulse energy, 200 kHz pulse rate and detection rate of 1-10%) giving spatial and chemical information on about $0.1–0.5 \, 10^9$ atoms per specimen. The analysis of atom probe tomography (APT) data was performed with the AP Suite by CAMECA and machine-learning-based methods as described in ref. 49.

Magnetic domain walls in the TEM specimens were imaged in magnetic-field-free conditions (Lorentz mode) using a spherical-aberration corrected transmission electron microscope operated at 300 kV. Fresnel defocus images were recorded using a direct electron counting 4k x 4k detector (Gatan K2 IS). The correlative chemical composition measurement was carried out using an electron probe-aberration corrected transmission electron microscope operated at 200 kV and equipped with an in-column energy dispersive X-ray spectroscopy (EDS) system. The images and spectra were processed using ThermoFisher Velox software. To perform off-axis electron holography, a voltage of 160 V was applied to the electrostatic biprism resulting in an interference fringe spacing of ~2.4 nm. Stacks of 10 electron holograms (each acquired for 4 s) were aligned and then averaged using the Holoview software to improve the signal to noise ratio of the reconstructed phase images. To isolate the magnetic contribution to the phase image ($\phi m$), the TEM lamella was physically flipped by 180° inside the TEM using a dedicated tomographic holder.

The reconstructed pairs of averaged phase images were digitally flipped, rotated and subtracted to isolate the $\phi m$. To create the magnetic induction maps, the $\phi m$ images underwent Gaussian smoothing and the cosine was taken to produce magnetic phase contours, and color wheels are used to show the direction of the projected induction.

Micromagnetic simulations were carried out by using the open-source GPU-accelerated finite-difference (FD) program Mumax3[50–52]. Starting from the idealized diamond structure based on refs. 19,26, a geometry model is created to consider additionally the 1:5-like interlayers and surface coating layers on the Zr-rich platelets. For the simulations, magnetic parameters based on the compositions determined by our APT measurements are used. Please refer to the supplemental material for details.

## Data availability
The datasets generated during and/or analysed during the current study are available by request to the corresponding authors.

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

## Acknowledgements

We acknowledge funding by the Deutsche Forschungsgemeinschaft (DFG, German Research Foundation), Project ID No. 405553726-CRC/TRR 270 DFG and by the German BMBF under the grant number 03XP0166A. NP is grateful for the funding for his scholarship by the IMPRS SURMAT. NP and BG are grateful for the funding of the Leibniz Prize 2020 by the DFG. L. M.-L acknowledges the European Research Council (ERC) "Horizon 2020" Program under Grant No. 805359-FOXON and Grant No. 957521-STARE. Authors Y.Y., D. O. and B.-X.X. appreciate their access to the Lichtenberg High-Performance Computer and the technique supports from the HHLR, Technical University of Darmstadt, and the GPU Cluster from the sub-project Z-INF of SFB/TRR 270. Y.Y. also highly thank the research assistant Eren Foya for performing intensive micromagnetic simulations. L.M.-L. acknowledges funding from the ERC "HORIZON.1.1" Program under Grant No. 101088712-ELECTRON and Grant No. 101189511-BED-TEM.

## Author contributions

The work we report is collegial and conceptionally the result of intense joint discussions across all co-authors, particularly during the CRC 270 retreats. S.G. performed magnetic measurements, S.E.M. and Kerr microscopy with support from IAR and additional measurements, including magnetic force microscopy (MFM), by F.M. on magnets processed by KÜ and MK. N.P. performed the APT experiments, N.P., A.S., C.F., and B.G. processed the data. E.A. performed the TEM, E.A. and L.M.L. interpreted the data. Y.Y. performed the micromagnetic

simulations, with input of DO and support by B.X.X. and H.Z.; A.K. and T.P.A. performed the magnetic imaging in the TEM and interpreted the data along with RDB.; K.P.S., S.G., E.A., L.M.L., Y.Y., B.X.X., A.K., T.P.A., N.P., O.G., B.G., drafted the manuscript following discussions of the results with all authors including M.F., K.D., L.A. and K.O.; All authors then contributed to the revisions.

## Funding

## Competing interests

The authors declare no competing interests.

## Additional information

[1]Institute of Materials Science, Technische Universität Darmstadt, Darmstadt, Germany. [2]Max Planck Institute for Sustainable Materials, Düsseldorf, Germany. [3]Advanced Electron Microscopy Division, Institute of Material Science, Technical University of Darmstadt, Darmstadt, Germany. [4]Ernst Ruska-Centre for Microscopy and Spectroscopy with Electrons, Forschungszentrum Jülich, Jülich, Germany. [5]SUPA, School of Physics and Astronomy, University of Glasgow, Glasgow, UK. [6]VACUUMSCHMELZE GmbH & Co. KG, Hanau, Germany. [7]Faculty of Physics and Center for Nanointegration (CENIDE), Universität Duisburg-Essen, Duisburg, Germany. [8]Department of Materials, Royal School of Mines, Imperial College London, London, UK. [9]Present address: Univ Rouen Normandie, CNRS, INSA Rouen Normandie, Groupe de Physique des Matériaux, Rouen, France. ✉e-mail: leopoldo.molina-luna@aem.tu-darmstadt.de; baptiste.gault1@univ-rouen.fr; konstantin.skokov@tu-darmstadt.de

