## [Peer review File · Nature Communications]

Identifying grain boundary and intragranular pinning centres in $\text{Sm}_2(\text{Co,Fe,Cu,Zr})_{17}$ permanent magnets to guide performance optimisation

Corresponding Author: Professor Baptiste Gault

Version 0:

Reviewer comments:

Reviewer #1

(Remarks to the Author)

The authors presented their sophisticated experimental and analysis work on coercivity of the Sm-Co based sintered magnets. The paper is well organized and written (few grammar mistakes like in line 244 should be removed).

My major concern is the logic on the analysis: the author gave a theme in the Introduction: "Common knowledge says that bulk magnets are only as strong as their weakest points". This is actually not true. Figure 1 has shown clearly that the coercivity value is determined by the strongest points (the high H_c regions). The weak H_c regions have only effect the squareness of the M-H loop.

This is a commonly accepted principle in permanent magnets. Another example is the hard-soft exchange-coupled composite magnets. If the overall coercivity is determined by the soft phase, no energy product can be produced.

Reviewer #2

(Remarks to the Author)

In this work, the authors performed multiscale microscopy and microanalysis on a bulk Sm-Co-based permanent magnets, in order to understand the negative impact of local weak points on performance of bulk magnets. Although the 2:17-type Sm-Co magnets are well-known and have wide applications in high-temperature environments, the present work did give new information, such as the changes in the composition and distribution of copper, and the atomic arrangements that enforce the pinning of magnetic domain walls. Together with the micromagnetic simulations, such results could be helpful for understanding of how multi-scale peculiarities change the hard magnetic properties of 2:17-type Sm-Co magnets.

I cannot fully understand the so-called "perfect defect" presented in the title. Defects, including point defects, dislocations, grain boundaries, and nanoprecipitates, coexist in the present magnets. I am wondering what type defects the authors focused on and what type defects played a dominant role on coercivity. "Defects" appear only in the title and abstract, but are rarely discussed in the main-text. To make the "perfect defect" solid, it is necessary to have a deep discussion.

The inhomogeneous cellular nanostructure in Fe-rich Sm-Co-based magnet (with Fe above 20wt.%, like the present composition) has been reported in the past. But it is not mentioned in the Introduction section.

More importantly, the local element segregation behavior in Fe-rich Sm-Co-based magnet (like that reported in Fig. 1a) has also extensively been reported. The local weak region (low coercivity area) with higher Sm content can have a finer cellular nanostructure than the strong region (high coercivity area) with lower Sm content. Besides, the Sm-rich region can also consist of only 2:17R phase (like the no. 2 region in Fig. S9), without any 1:5 or Z platelets to pin the magnetic domain walls (e.g. in *Materials* 22 (2022) 101382). Such region free of 1:5 or Z platelets (precipitate-free zones, PFZs) loses [001] texture locally, also being harmful for remanence. Beside this region, such as no. 3 region in Fig. S9, the local segregations of Cu and Zr could form the mixture of 1:3, 2:7 and 5:19 (Zr-stabilized $\text{Sm}_{n+1}\text{Co}_{5n-1}$, like reported in *Journal of Materials Science & Technology* 85 (2021) 56–61) phases, which are also harmful for magnetic properties (both coercivity and remanence). In my opinion, the local low coercivity area identified by TEM/APT in this work is just one of the local weak areas, while the

PFZs (that could be the weakest area) and $\text{Sm}_{n+1}\text{Co}_5\text{Zr}_{n-1}$ phases are ignored.

It is appreciated that the detailed microstructural and compositional investigations between the finer cells area and the coarser cells area can give rise to a deeper understanding. In this case, the finer cells area could be deemed “defect” (in a more common sense) for this magnet. But, the present work lacks an important step, “how to engineer the defect” or “eliminate the defect towards larger coercivity”. This makes the present work incomplete or the title “engineering” unsuitable. Answer to this question has been given in the literature, such as the optimization of solution-treatment processing.

Some other issues:

1. “Pinning” usually refers to the “pinning of magnetic domain walls” not “pinning of magnetic domains”.
2. Please give the special range of δ in $\text{Sm}(\text{CoFeCuZr})_{7\pm\delta}$. Line 53, page 3.
3. $\text{Sm}(\text{CoFeCuZr})_{7\pm\delta}$ magnets can contain 2:17H metastable or 2:17R equilibrium phases. It's better to indicate that the mentioned 2:17 phase in the magnet is 2:17R.
4. Please indicate c-axis in Figs. 1b-d. This will help the colleagues to understand along which direction the domain configurations were taken.
5. The phrase “wetting layer” could be reconsidered. Both 1:5 precipitates and Z platelets are formed by solid-state decomposition, there is no liquid phase during their nucleation and growth. “Transient diffusion layer” could be a better choice.

Version 2:

Reviewer comments:

Reviewer #2

(Remarks to the Author)

The authors have made significant efforts to revise the manuscript, with most concerns have been addressed. I suggest to accept it after addressing the following minor issues.

1. “During sintering, it is essential to have a phase with a lower melting point, which melts at the sintering temperature, filling the gaps between solid grains, enhancing diffusion and promoting densification. As a result, the final magnet exhibits slight compositional differences between the grain interiors and the regions near grain boundaries (GBs).”

This statement should be reconsidered. There is no liquid phase during sintering of SmCoFeCuZr magnets, unlike NdFeB -based magnets. The compositional difference between grain interiors and grain boundaries is due to solid-state diffusion and grain growth during sintering and high temperature homogenization processes (since the particle size prior to sintering is much smaller than the grain size), in particular for the magnets with high Fe content.

2. “Although the GB region experiences magnetization reversal at relatively low fields (0.1-0.3 T), this reversal is limited and confined to the thin outer layer of the grains, and does not have a significant effect on the magnet's overall coercivity”. The present results seem to support this statement. However, the weak GB region leads to poor squareness factor of the demagnetization curve, hence being harmful for achieving large maximum energy product, the key figure of merit of hard magnets. I suggest to their effects on maximum energy product.

Reviewer #3

(Remarks to the Author)

This manuscript comprehensively investigates the microstructure, phase composition, magnetic structure, and coercivity mechanism of $\text{Sm}_2(\text{CoFeCuZr})_{17}$ magnets with two types of permanent magnetic properties and finds that the optimal microstructure within the grain itself is crucial for achieving the desired magnetic properties. However, I think the manuscript could be further optimized. Please consider the following questions:

1. The title is not very precise. The authors have forcibly elevated the research content to the level of engineering, and they mainly reveal the structure of magnets with different properties without proposing engineering measures and methods for controlling this structure.
2. Why is it necessary to cite references in the abstract?
3. In line 70, you mention that Sintered $\text{Sm}(\text{CoFeCuZr})_{7\pm\delta}$ ($\delta=0.1\dots 1$) magnets consist of grains of about 100 μm in diameter. May I ask if it is 100 μm or 100nm?
4. In lines 105-109, why is there a difference in performance if two magnets come from the same industrial batch and the raw materials and production process are exactly the same? Therefore, does this mean that production parameters were not the same? For example, during the preparation process, the samples come from different locations inside the heat treatment furnace, which means that the actual heat treatment temperature of the samples is different. Should you first trace the reasons for the differences in performance?
5. In lines 130-132, Figure 1a is from magnet A, and Figures 1b-d are from sample B. I think the author made a mistake. Should it be sample A? The $\text{Sm}(\text{Co, Cu, Fe, Zr})_z$ magnet is composed of $\text{Sm}_2(\text{Co, Fe})_{17}$ cells with a diameter of about 100nm and $\text{Sm}(\text{Co, Cu})_5$ cell walls with a thickness of about 10nm. The grains circled by red lines in Figure 1a (about 20 microns) also contain about 200 cellular structures along a unidirectional length. Therefore, the grains in Figure 1a are not the actual grains in sintered magnets.

6. The authors only focused on two areas close to grain boundaries. Are other phenomena, such as the different magnetic domain structures in large grains and the varying sizes, contents, and distributions of oxides in various grains, as shown in Figure 1b, not important?
7. Two areas close to grain boundaries, outlined in red, appear brighter. Is it because the cellular structure in these regions is smaller and there are relatively more grain boundaries, resulting in a higher content of $\text{Sm}(\text{Co}, \text{Cu})_5$ phase?
8. In Figure S1, (1) the authors believe that the remanent polarization of two magnets is almost the same, which is questionable. This difference has always existed during the magnetization and demagnetization processes, and it is not negligible, indicating that there may be differences in the composition of the two magnets. (2) Magnet B exhibits a more pronounced step than magnet A under a tiny demagnetization field. The nucleation field (HN) of the phase should be smaller than the HN of the same phase in magnet A, right? (3) Fig. S1 shows the Initial curve and demagnetization of samples A and B after applying a field of 14 T. Are the B_s of two magnets the same under a 14T magnetic field? Why does Figure S1 only show the curve under a 5T magnetic field? The complete hysteresis loop should be displayed, which may provide more information. (4) The authors point out that a shoulder appears for sample A at demagnetizing fields as small as 0.3 T. Please indicate this point in the figure. It seems that -0.3T corresponds to sample B instead of sample A, right? (5) Sample A shows a two-step demagnetization with increased susceptibilities in demagnetizing fields larger than approximately 1.0 T. How did you observe the susceptibility during the demagnetization process?
9. In Figure S2, (1) on line 39, the authors present Fig. c for sample A has an average grain size of 74 μm , whereas the grain size in (sample?) B is 200 μm . However, Table S1 shows that the average grain size of sample A is 200 μm , while sample B's is 220 μm , which is inconsistent between the previous and subsequent descriptions. Furthermore, such results cannot be seen from Figures S2c and f. On the contrary, Figure C has larger grain sizes and fewer grain boundaries. (2) Did the authors notice the difference in magnetic domain structure between Figure S2b and e? Have you noticed the differences in size and distribution of those white particles? Have you noticed the differences in magnetic domain structure at grain boundary positions? (3) Did the authors notice the white line in Figure S2f? (4) Should magnetic domain structure images be provided after applying different demagnetization fields? For example, can you provide the magnetic domain images near the inflection points on the demagnetization curve?
10. The data source for Table S1 was not clearly explained. The cell size in Table S1 is close to 300nm, significantly different from the 100nm described in line 73. What is the reason for this?
11. Does the smaller the cell size, the smaller the coercivity? Can the composition measured by SEM-EDS in Table S2 genuinely reflect the composition of the cellular grains for such a small size? What does the precipitate in Table S2 refer to? What is its size?
12. Does Fig. S4 show that the smaller the magnetic domain size, the lower the coercivity?
13. Are Figures 1c-d SEM images instead of Kerr microscopy images? Why can't we see magnetic domains or magnetic domain walls? Can you provide magnetic domain images under different demagnetization fields (such as -0.3T, -1.0T) to prove that dark regions are demagnetized first? Can you accurately provide the nucleation field for demagnetizing these dark grains?
14. Comparing Figure 1 and Figure 2, it can be found that there are many defects in sample A, including black particles in the sample and large black particles on the grain boundaries and inside the grains, and the microstructure and magnetic domain structure inside different grains are also different. Does this mean there are significant differences in the two samples' intrinsic properties, such as composition and microstructure?
15. According to Figure 1 and Figure S2, the low coercivity region mentioned by the authors should be located near the grain boundary, which is several tens of micrometers. However, Figure 3a has a size of less than 2 μm . How did the authors distinguish between low and high coercivity regions in TEM? Is it based on the cell size? Figure 1 shows regions with different cell sizes should be within the same large grain.
16. The authors define different regions in sample A as low-coercivity regions and high-coercivity regions. Still, the highest coercivity of the magnet is 2.2T, so the high coercivity regions in sample A are also defective regions.
17. Is the density of Z-Platelets in Figure 4a higher than in Figure 3? Is the morphology of each Z-Platelet group also different from that in Figure 3?
18. On line 254, APT confirms the observations by SEM in Fig. 1: the low-Hc region is richer in Sm and Cu by approximately 1 at.% and poorer in Fe and Co by 1–2 at.% compared to the high-Hc region. If there were only such a slight difference in composition, would the anisotropic fields of these phases and the pinning resistance between them differ so much?
19. The infocus image in Fig. S7a shows an approximately 1 μm wide region between the high-Hc and low-Hc grain. Why can't such a wide grain boundary have an adverse effect on the coercivity observed in sample A? Why is poor performance necessarily due to the cellular structure within the grains? From Figure S8, it can be seen that the composition within the grain boundaries is also uneven. For example, there are regions rich in Sm-Cu-Zr and poor in Co-Fe in the grain boundaries. Why should these regions not have a low nucleation field? The magnetization curve (Figure 9a) reveals easy demagnetization in the GB region.
20. There are many errors in the references.
21. The writing format also does not meet the journal's requirements.

Version 3:

Reviewer comments:

Reviewer #2

(Remarks to the Author)

My concerns have been addressed.

Reviewer #3

(Remarks to the Author)

The reviewer has answered my concerns well, but there are still some issues with the standardization of the writing, such as:

1. The number $\text{Sm}_2(\text{Co}, \text{Fe}, \text{Cu}, \text{Zr})_{17}$ in the title should be subscripted.
2. The chemical formula writing is confusing, with examples such as $\text{Sm}(\text{Co}, \text{Fe}, \text{Cu}, \text{Zr})_7$ and $\text{Sm}(\text{CoFeCuZr})_7$ appearing throughout the text, some with commas and some without.
3. On page 16, there is still an error (Error! Bookmark not defined) that has not been corrected.
4. The writing of supplementary materials is very non-standard, such as: (1) The font size in the supplementary materials is inconsistent. (2) The font size in the supplementary materials is inconsistent, such as in Table S3. (3) The tables in the supplementary materials are not standardized. (4) The formula writing and interrupt sign writing in the supplementary materials are not standardized.

Reviewers' comments are reported below, and our answers are in blue.

Reviewer #1 (Remarks to the Author):

The authors presented their sophisticated experimental and analysis work on coercivity of the Sm-Co based sintered magnets. The paper is well organized and written (few grammar mistakes like in line 244 should be removed).

My major concern is the logic on the analysis: the author gave a theme in the Introduction: "Common knowledge says that bulk magnets are only as strong as their weakest points". This is actually not true. Figure 1 has shown clearly that the coercivity value is determined by the strongest points (the high Hc regions). The weak Hc regions have only effect the squareness of the M-H loop.

This is a commonly accepted principle in permanent magnets. Another example is the hard-soft exchange-coupled composite magnets. If the overall coercivity is determined by the soft phase, no energy product can be produced.

We are very grateful to the reviewer for their assessment of our work and we must thank them for the criticism, since it motivated us to improve the clarity of our manuscript. In particular, the major concern of the reviewer about the wording we had chosen has been addressed and these sections of the manuscript have been completely rewritten (sections in blue in our manuscript).

The problem of a 'weak link' in a permanent magnet, consisting of several types of areas with different coercivity along with the complex microstructure in each of them, is very intriguing because its solution provides a potential opportunity to improve the functional properties of the magnet by avoiding the appearance of this 'weak link'.

To address this more specifically, we have performed an additional complete study on an optimally annealed reference sample. This sample has the highest coercivity we were able to obtain for this composition and does not contain any regions with reduced coercivity. The results are all integrated in the new version of the manuscript now and discussed in light of the initial set of results and the discussion on the 'weak link', as detailed in the following.

A new figure, Fig S2 below, shows that the coercivity of the reference sample without middle-coercivity areas near the grain boundaries $H_c \sim 3$ T, whereas the sample with 'weak points' shows the net coercivity of 2.2 T. It turned out to be very important that demagnetization of the regions adjacent to the grain boundary in fields of 0.2–0.3 T is observed in both samples, which is evident from the new Figure 2 added to the manuscript. However, this process does not develop across the entire magnet and, therefore, such regions that exhibit a low coercivity, but their presence is not critical for the high-coercive state of the entire magnet.

Although there are many conjectures across the literature that regions near grain boundaries are a weak links, we demonstrate that, conversely to the common belief, they are not critical to the overall coercivity of the bulk magnet. In a new figure, Fig S3 below, we demonstrate that the microstructure near the grain boundary is effectively different to that of the bulk of the grains.

Fig S2: SEM images using BSE ((a), (d) and (g)) and Kerr microscopy images of the thermally demagnetized state ((b), (e) and (h)) and after applying an external field of 0.8 T ((c), (f) and (i)) to the saturated samples A (top row), B (bottom row), respectively. The same sample regions are shown for SEM and Kerr microscopy.

Fig. S3 HR-SEM image of A sample of a high H_c region on the left side of the grain boundary (marked by green lines) and a low H_c region on the right. A finer cell structure is visible in the low H_c region. The surface is perpendicular to the c -axis of the 2:17 phase.

In light of these considerations, the logic of our analysis is to reveal the nanostructure of the areas with low coercivity, by using the same suite of microstructural characterization techniques that we had initially used. For instance a new figure, Fig 7, shows the APT analysis of the corresponding region, demonstrating the difference in composition and microstructure near the GB. Additional APT and (HR)TEM from the bulk of the high performing sample allow for a comparative study, demonstrating the critical role of a defect-state consisting of a thin 1:5 layer at the interface of the Z-phase. By using direct observation of domain wall pinning along with micromagnetic modelling, we provide an understanding why such nanostructure is responsible for the reduction of the whole magnet performance. The additional points in the discussion are highlighted in blue.

Figure 7: APT data for the grain boundary (GB) of the reference sample B. a 3D reconstruction with Cu-rich isosurface in blue (isoconcentration value 5 at.% Cu) and blue/green points representing Cu/Zr atoms, showing features of incomplete cellular (“1:5 phase”, #1) and lamellar structure (“Z-phase”, #2) and a Sm, Zr, and Cu-enriched particle (#3) with corresponding 1D profiles (b,c,d) across the respective features as indicated by named orange arrows.

Figure 8: HAADF STEM images of (a) low- and (b) high-Hc areas of the A magnet. Areas are labelled according to their atomic stacking being matched with theoretical structures (cp. Fig. S5). (c, d) Magnified images highlighting the existence of 1:5 phase as interlayers and coating layers at Z-platelets. (e) Image intensity profiles extracted from dashed lines with corresponding colour in (d) show the mixed nature of the coating layer. (f) Under- and overfocused Fresnel images of magnetic domain walls in low-Hc grain. The defocus applied was 0.5 μm . (g, h) An amplitude image and the corresponding magnetic induction map showing the pinning of domain walls at the phase boundaries of the 1:5 and 2:17 phases. Solid and dashed lines mark the domain wall and 1:5 phase locations, respectively. The colours and contours lines indicate the field line direction and strength. The contour spacing is 2π radian.

Reviewer #2 (Remarks to the Author):

In this work, the authors performed multiscale microscopy and microanalysis on a bulk Sm-Co-based permanent magnets, in order to understand the negative impact of local weak points on performance of bulk magnets. Although the 2:17-type Sm-Co magnets are well-known and have wide applications in high-temperature environments, the present work did give new information, such as the changes in the composition and distribution of copper, and the atomic arrangements that enforce the pinning of magnetic domain walls. Together with the micromagnetic simulations, such results could be helpful for understanding of how multi-scale peculiarities change the hard magnetic properties of 2:17-type Sm-Co magnets.

I cannot fully understand the so-called “perfect defect” presented in the title. Defects, including point defects, dislocations, grain boundaries, and nanoprecipitates, coexist in the present magnets. I am wondering what type defects the authors focused on and what type defects played a dominant role on coercivity. “Defects” appear only in the title and abstract, but are rarely discussed in the main-text. To make the “perfect defect” solid, it is necessary to have a deep discussion.

We thank the reviewer for this comment. We accept that the concept of the “perfect defect” may be too obscure – it follows on many discussions that have been going around our large group of authors but may not be clear outside of this group.

We have hence removed mentions of this concept, from the title, that now reads:

Towards engineering the perfect pinning centers in high-performing $\text{Sm}_2(\text{CoFeCuZr})_{17}$ -type permanent magnets

And across the entire manuscript.

The inhomogeneous cellular nanostructure in Fe-rich Sm-Co-based magnet (with Fe above 20wt.%, like the present composition) has been reported in the past. But it is not mentioned in the Introduction section.

We are grateful to the reviewer for this comment and have made the necessary changes to the text in accordance with his wishes. This is now in the introduction, with more details on the previous work in this area. The introduction now contains this statement:

The production of Sm-Co-Fe-Cu-Zr magnets involves a complex metallurgical process, that includes milling, compaction in magnetic field, sintering, high-temperature homogenization, isothermal aging, slow cooling, and second-step aging. During sintering, it is essential to have a phase with a lower melting point, which melts at the sintering temperature, filling the gaps between solid grains, enhancing diffusion and promoting densification. As a result, the final magnet exhibits slight compositional differences between the grain interiors and the regions near grain boundaries (GBs)^{22,29}. The GB area is Cu-depleted and includes magnetically softer phases (e.g. $\text{Sm}_{n+1}\text{Co}_5n-1$), precipitate-free-zones and other defects and microstructural features^{30,31}. Near the GBs of sintered magnet, the 1:5/2:17 cellular nanostructure is coarse and inhomogeneous, and this fact have been recognized as important microstructural origins for the lower-than-ideal energy product of precipitate-hardening Sm-Co-Fe-Cu-Zr magnets with high Fe content^{24,27,28,32}. In addition, magnetization reversal in the GB region occurs in relatively weak fields (0.1–0.3 T), causing a distinctive kink in the hysteresis loop, while the rest of the sample demagnetizes at much stronger fields, closer to H_c (2–3 T) ^{24,27,28,32}. This is why GBs are often recognized as "weak links" in terms of coercivity. However, a detailed study reporting the complete evolution of the demagnetization

process, in relation to the characteristic microstructure and composition of defects is missing, leaving a knowledge gap regarding their true influences the final coercivity and energy product and hindering more precise microstructural design of permanent magnets.

More importantly, the local element segregation behavior in Fe-rich Sm-Co-based magnet (like that reported in Fig. 1a) has also extensively been reported. The local weak region (low coercivity area) with higher Sm content can have a finer cellular nanostructure than the strong region (high coercivity area) with lower Sm content. Besides, the Sm-rich region can also consist of only 2:17R phase (like the no. 2 region in Fig. S9), without any 1:5 or Z platelets to pin the magnetic domain walls (e.g. in *Materialia* 22 (2022) 101382). Such region free of 1:5 or Z platelets (precipitate-free zones, PFZs) loses [001] texture locally, also being harmful for remanence. Beside this region, such as no. 3 region in Fig. S9, the local segregations of Cu and Zr could form the mixture of 1:3, 2:7 and 5:19 (Zr-stabilized $\text{Sm}_{n+1}\text{Co}_5\text{n-1}$, like reported in *Journal of Materials Science & Technology* 85 (2021) 56–61) phases, which are also harmful for magnetic properties (both coercivity and remanence). In my opinion, the local low coercivity area identified by TEM/APT in this work is just one of the local weak areas, while the PFZs (that could be the weakest area) and $\text{Sm}_{n+1}\text{Co}_5\text{n-1}$ phases are ignored.

We are grateful to the reviewer for raising this point. We have now expanded on these aspects, added references and added much discussion, based on the additional comparison with the new reference sample fabricated and tested. As discussed above, we could demonstrate that although grain boundaries are indeed a weak area, their influence remains localized, whereas the newly identified defective phases affect more deeply the properties of the magnet. This is discussed in detail in the manuscript now, as detailed above and in the following:

In sample B, APT analyses were performed in the center of the grain and at a GB, with representative reconstructions shown in Fig. 6 and Fig. 7, respectively. The grain center exhibits a fully developed nanoscale structure with Fe-rich 2:17 phase cells, Cu-rich 1:5 phase cell walls, and Zr-rich Z-platelets, Fig. 6a, highly similar to the high H_c areas of the A sample: (1) The ~10 nm thick 1:5 phase shows a high Cu concentration peaking at ~30 at.%, as shown by 2D-Cu profile and 1D profile in Fig. 6b and 6c; (2) Z-platelets have discontinuous Cu-rich coating layers peaking at 8 at.% Cu (Fig. 6d), with thicker platelets ~10nm exhibiting Cu-enriched centers (Fig. 6e), that are not found in the thinner platelets ~ 5 nm (Fig. S6 in the supplementary material).

The GB is primarily Fe-rich 2:17 phase without cellular or lamellar structure (Fig. 7). At a low isosurface value of 5 at.% Cu, features resembling residual cellular and lamellar structure and a Sm, Zr, and Cu-enriched particle become apparent as seen in Fig. 7a. Two parallel planar and discontinuous features, resembling the Z-phase, exhibit a Cu-rich coating layers, compatible to Z-platelets in the grain center, along with Zr and Co composition variations, as seen in a representative 1D-profile of one “Z-platelet” (Fig. 7b). Similar features were observed in the GB of the sample A by scanning-TEM (STEM)-EDS (Fig. S8). Another set of 8 nm thick planar features forms a junction, resembling the 1:5 phase with low Cu-content peaking at 8 at.%, as a representative 1D profile in Fig. 5c shows. Additionally, a ~100 nm large particle enriched in Sm, Zr, and Cu with a composition $\text{Sm}_{16}\text{Co}_{56}\text{Fe}_{12}\text{Zr}_7\text{Cu}_8$ was partially captured (Fig. 7d), inside which appears a set of Zr-rich and Cu-rich layers around 5-10 nm thick, consistent with STEM-EDS results in the GB of the sample A (Fig. S8).

Figure 6: APT data for the grain centre of the reference sample B. **(a)** 3D reconstruction showing the geometrical distribution of the 1:5 phase (blue, isoconcentration value 12 at.% Cu) and the Z-phase (green, isoconcentration value 6 at.% Zr). Blue and green dashed lined boxes indicate a representative Cu-2D distribution for the 1:5 phase-plates **(b)** and a rotated 3D reconstruction of the Z-platelets **(d)**, respectively. 1D composition profiles are measured perpendicular to the representative 1:5 phase-plates **(c)** and the Z-platelets **(e)** as indicated by the orange arrows and crossed circles in **a,b,d**. The nanoscale cellular lamellar microstructure in the grain centre of the sample B highly resembles that in the high H_c area in the A sample.

Grain boundary

Figure 7: APT data for the grain boundary (GB) of the reference sample B. **a** 3D reconstruction with Cu-rich isosurface in blue (isoconcentration value 5 at.% Cu) and blue/green points representing Cu/Zr atoms, showing features of incomplete cellular (“1:5 phase”, #1) and lamellar structure (“Z-phase”, #2) and a Sm, Zr, and Cu-enriched particle (#3) with corresponding 1D profiles (**b,c,d**) across the respective features as indicated by named orange arrows.

It is appreciated that the detailed microstructural and compositional investigations between the finer cells area and the coarser cells area can give rise to a deeper understanding. In this case, the finer cells area could be deemed “defect” (in a more common sense) for this magnet. But, the present work lacks an important step, “how to engineer the defect” or “eliminate the defect towards larger coercivity”. This makes the present work incomplete or the title “engineering” unsuitable. Answer to this question has been given in the literature, such as the optimization of solution-treatment processing.

Our intention was to provide a reflection point that can be used to engineer hard magnets – with a particular emphasis on the role of microstructure and defects. However, if such an interpretation raises some questions, we have changed the discussion in light also of the new results added to the manuscript, including the aspect of “engineering the defect”. In the conclusion, we have added this section:

Achieving these improvements will likely require adjustments to alloy composition and heat treatment procedures. If the 1:5 phase interlayers within the Z-platelets also act as additional pinning centers—a hypothesis that requires further investigation—similar engineering strategies could be applied to these interlayers to further enhance coercivity in 2:17-type magnets. The presence of low-H_c regions with suboptimal nanoscale structure and microchemistry can hence be considered a “weak link” of the magnet (Fig. S12a). In addition, contrary to the common belief that the grain boundaries are the “weakest link” in bulk magnets, we have shown that this is not the case. Indeed, the GB region undergoes magnetization reversal in weak fields, (0.1-0.3 T) but the process stops at the boundary of the grain interior, where domain walls meet the nanostructure of the grain and get pinned. This magnetization

reversal at GBs does not affect the overall coercivity of the magnet, manifesting itself as a small kink in low negative field in the hysteresis loop (see Fig. S1). Our work shows that it is the optimal microstructure inside the grain that is the key to optimal magnet properties. Our study highlights the importance of complete identification of each building block constituting these permanent magnets at multiple length scales, and understanding the subtle differences in their arrangements, composition and physical properties. All of this is indispensable for the further improvement of unique characteristics such highly sought-after permanent magnets.

We hope that these are now sufficient in the view of the reviewer.

Some other issues:

1. “Pinning” usually refers to the “pinning of magnetic domain walls” not “pinning of magnetic domains”.
2. Please give the special range of δ in $\text{Sm}(\text{CoFeCuZr})_{7\pm\delta}$. Line 53, page 3.
3. $\text{Sm}(\text{CoFeCuZr})_{7\pm\delta}$ magnets can contain 2:17H metastable or 2:17R equilibrium phases. It’s better to indicate that the mentioned 2:17 phase in the magnet is 2:17R.
4. Please indicate c-axis in Figs. 1b-d. This will help the colleagues to understand along which direction the domain configurations were taken.
5. The phrase “wetting layer” could be reconsidered. Both 1:5 precipitates and Z platelets are formed by solid-state decomposition, there is no liquid phase during their nucleation and growth. “Transient diffusion layer” could be a better choice.

All these points have been addressed in the revised version of the manuscript. All changes are marked in blue in the manuscript.

Dear Editor,

We would like to thank you and the reviewers for the careful evaluation of our manuscript and for the constructive remarks that helped us to improve the quality and clarity of the work. We have carefully considered all the comments and revised the manuscript accordingly. Below, we provide a detailed response to each point raised by the reviewers, indicating the changes made in the text (in blue) and, where necessary, providing additional explanations to clarify our results and conclusions.

Reviewer #2 (Remarks to the Author):

The authors have made significant efforts to revise the manuscript, with most concerns have been addressed. I suggest to accept it after addressing the following minor issues.

1. *“During sintering, it is essential to have a phase with a lower melting point, which melts at the sintering temperature, filling the gaps between solid grains, enhancing diffusion and promoting densification. As a result, the final magnet exhibits slight compositional differences between the grain interiors and the regions near grain boundaries (GBs).”*

This statement should be reconsidered. There is no liquid phase during sintering of SmCoFeCuZr magnets, unlike NdFeB-based magnets. The compositional difference between grain interiors and grain boundaries is due to solid-state diffusion and grain growth during sintering and high temperature homogenization processes (since the particle size prior to sintering is much smaller than the grain size), in particular for the magnets with high Fe content.

The Vacuumschmelze company prepared the samples, and the persons in charge of this production process confirmed that there is a significant amount of liquid phase during the initial sintering Sm₂Co₁₇ magnets. That also explains why the grain size of such magnets increases from about 7 μm for the powder particles to more than 100 μm for the magnet. After the liquid phase sintering at about 1200 °C the temperature is lowered to about 1170°C. At this temperature, all the liquid phase solidifies, and the homogenization of the remaining compositional differences that have developed during the sintering phase must be accomplished by solid-state diffusion.

The manuscript was modified to make our argument more precise:

During sintering of Sm–Zr–Co–Cu–Fe magnets, compositional heterogeneity develops between grain interiors and grain boundaries as a consequence of solid-state diffusion and grain growth. In particular, the redistribution of transition metal elements is strongly influenced by Fe content: higher Fe concentrations enhance diffusion kinetics and promote preferential segregation at grain boundaries, thereby accentuating the difference in chemical composition between bulk grains and boundary regions.

2. *“Although the GB region experiences magnetization reversal at relatively low fields (0.1-0.3 T), this reversal is limited and confined to the thin outer layer of the grains, and does not have a significant effect on the magnet's overall coercivity”.*

The present results seem to support this statement. However, the weak GB region leads to poor squareness factor of the demagnetization curve, hence being harmful for achieving

large maximum energy product, the key figure of merit of hard magnets. I suggest to their effects on maximum energy product.

Thank you for this valuable comment. We examined an optimally processed magnet (magnet B), which exhibited very good squareness of the hysteresis loop despite the presence of an intergranular phase with relatively low coercivity. In contrast, for a non-optimally processed magnet A, regions with non-ideal microstructure were observed, leading to pronounced kinks in the hysteresis loop at low negative fields and, consequently, a deterioration of the magnetic performance.

To clarify this point in the manuscript, we have added the following text:

Nevertheless, even a thin GB region can slightly reduce the squareness factor of the demagnetization curve and thereby lower the maximum energy product. However, since such regions are unavoidable due to the intrinsic limitations of the technological process in magnets A and B, the resulting reduction in properties from the GB phase is relatively minor.

Reviewer #3 (Remarks to the Author):

This manuscript comprehensively investigates the microstructure, phase composition, magnetic structure, and coercivity mechanism of $\text{Sm}_2(\text{CoFeCuZr})_{17}$ magnets with two types of permanent magnetic properties and finds that the optimal microstructure within the grain itself is crucial for achieving the desired magnetic properties. However, I think the manuscript could be further optimized. Please consider the following questions:

1. The title is not very precise. The authors have forcibly elevated the research content to the level of engineering, and they mainly reveal the structure of magnets with different properties without proposing engineering measures and methods for controlling this structure.

We changed the title of the manuscript to make it more scientific than engineering.

Understanding grain boundary and intragrain pinning centers in $\text{Sm}_2(\text{Co,Fe,Cu,Zr})_{17}$ permanent magnets: mapping the path to high performance

2. Why is it necessary to cite references in the abstract?

We removed citations from the abstract, this was before we transferred from Nature to Nature Communications in the very first version of the paper.

3. In line 70, you mention that Sintered $\text{Sm}(\text{CoFeCuZr})_{7\pm\delta}$ ($\delta=0.1\dots1$) magnets consist of grains of about $100\ \mu\text{m}$ in diameter. May I ask if it is $100\ \mu\text{m}$ or 100nm ?

It is true, the size of the grains is $100\ \mu\text{m}$, but the typical size of the cellular nanostructure inside of the grains is 50-100 nm.

4. In lines 105-109, why is there a difference in performance if two magnets come from the same industrial batch and the raw materials and production process are exactly the same? Therefore, does this mean that production parameters were not the same? For example, during the preparation process, the samples come from different locations inside the heat

treatment furnace, which means that the actual heat treatment temperature of the samples is different. Should you first trace the reasons for the differences in performance?

We added sentences to address this question in the text of the manuscript:

All production parameters were identical, except for the high-temperature homogenization step: magnet A with lower coercivity, was annealed at a temperature 7 K higher than magnet B, produced under optimal conditions. Given that such temperature deviations can occur in large-scale technological processes, it is essential to understand the corresponding microstructural changes that lead to a reduction in coercivity.

Also, we add this text to the 'methods' section:

The only difference between the two studied magnets was the high-temperature homogenization step: magnet A, which exhibited lower coercivity, was annealed at a temperature 7 K higher than magnet B, processed under optimal conditions.

5. In lines 130-132, Figure 1a is from magnet A, and Figures 1b-d are from sample B. I think the author made a mistake. Should it be sample A? The $\text{Sm}(\text{Co}, \text{Cu}, \text{Fe}, \text{Zr})_z$ magnet is composed of $\text{Sm}_2(\text{Co}, \text{Fe})_{17}$ cells with a diameter of about 100nm and $\text{Sm}(\text{Co}, \text{Cu})_5$ cell walls with a thickness of about 10nm. The grains circled by red lines in Figure 1a (about 20 microns) also contain about 200 cellular structures along a unidirectional length. Therefore, the grains in Figure 1a are not the actual grains in sintered magnets.

We are absolutely sure that all the information in Figure 1 corresponds to sample A, and what is shown in Figure 2 corresponds to sample B. The red line does not mark the grain, but the region inside the grain with reduced coercivity. Thus, we show that it is precisely within the grain that there may be regions with reduced coercivity, which reduce the overall performance of the magnet. We changed this phrase for better clarity:

Backscattered electron (BSE) scanning micrograph in Fig. 1a, shows multiple grains of the A magnet, and we focus here on two areas close to grain boundaries inside of the grains, which outlined in red, appear brighter, indicative of a higher average atomic weight, i.e. a higher Sm content.

6. The authors only focused on two areas close to grain boundaries. Are other phenomena, such as the different magnetic domain structures in large grains and the varying sizes, contents, and distributions of oxides in various grains, as shown in Figure 1b, not important?

With the exception of the low-coercivity regions of magnet A, the microstructure of the remaining parts of the magnets is completely identical. For example, lines 314–318 in the manuscript describe this:

TEM and APT observations reveal that the nanoscale structure of the grain interior in magnet B closely resembles that of the high H_c region in magnet A. The main distinction between the samples A and B lies in the presence of low H_c areas in magnet A, consistent with SEM and Kerr results (Figs. 1 and 2). Magnet B thus can be considered consisting of grains with high H_c nanoscale structure.

Also in the supplementary materials, in Tables S2 and S3, a detailed and exhaustive description of the various sections of the microstructure of both samples is presented.

7. Two areas close to grain boundaries, outlined in red, appear brighter. Is it because the cellular structure in these regions is smaller and there are relatively more grain boundaries, resulting in a higher content of $\text{Sm}(\text{Co}, \text{Cu})_5$ phase?

The low-Hc region is richer in Sm and Cu by approximately 1 at.% and poorer in Fe and Co by 1–2 at.% compared to the high-Hc region. We discuss this in several places in the manuscript, for example, lines 254-257 in the manuscript:

APT confirms the observations by SEM in Fig. 1: the low-Hc region is richer in Sm and Cu by approximately 1 at.% and poorer in Fe and Co by 1–2 at.% compared to the high-Hc region. Supporting measurements by SEM-EDS can be found in Tab. S3.

And, for example, lines 258-274 in the manuscript:

Compared to the high-Hc region, in the low-Hc region, the 1:5 phase is closer to an ideal diamond shaped structure, with a relatively denser network, with 8% more of the 1:5 phase and 5% more of the Z-phase (Tab. S3). The composition of the 2:17 phase is compatible within the two areas, see Tab. S3. In contrast, across representative 1:5-phase plates (dashed blue boxes in Figs. 5a1 and 5b1), profiles (yellow arrows in Figs. 5a2, 5b2) reveal a composition lower by 5 at.% in Cu and higher by 4 at.% in Co in the low-Hc region. Composition profiles through representative Z-platelets (dashed green boxes in Figs. 3a1 and 3b1), plotted in Figs. 5a4, 5b4 reveal two striking differences between the high-Hc and low-Hc areas: the Cu concentration increases in the centre of the Z-platelet by up to 4 at.% and the Z-2:17 phase boundary is enriched up to 8 at.% Cu in a ~5 nm thin region for the high-Hc area. A two-dimensional view in Fig. 5b5 suggests that this Cu-enrichment stems from a Cu-rich coating layer which is discontinuous. A similar increase in Cu at the Z-platelet–2:17 boundary is reported in Ref.40, and a lower but detectable increase in Cu at the Z-platelet interface was also observed in Ref.41, but neither are discussed any further. Overall, the high-Hc region of the magnet A contains less of a 1:5-phase that is richer in Cu, and has fewer Z-platelets, but also with increased Cu concentrations, both inside the platelets and at the interfaces to the 2:17 phase.

Also, tables S2 and S3 contains detailed chemical composition of all areas of the samples, including the region outlined the red line.

8. In Figure S1, (1) the authors believe that the remanent polarization of two magnets is almost the same, which is questionable. This difference has always existed during the magnetization and demagnetization processes, and it is not negligible, indicating that there may be differences in the composition of the two magnets. (2) Magnet B exhibits a more pronounced step than magnet A under a tiny demagnetization field. The nucleation field (HN) of the phase should be smaller than the HN of the same phase in magnet A, right? (3) Fig. S1 shows the Initial curve and demagnetization of samples A and B after applying a field of 14 T. Are the Bs of two magnets the same under a 14T magnetic field? Why does Figure S1 only show the curve under a 5T magnetic field? The complete hysteresis loop should be displayed, which may provide more information. (4) The authors point out that a shoulder appears for sample A at demagnetizing fields as small as 0.3 T. Please indicate this point in the figure. It seems that -0.3T corresponds to sample B instead of sample A, right? (5) Sample A shows a two-step demagnetization with increased susceptibilities in demagnetizing fields larger than approximately 1.0 T. How did you observe the susceptibility during the demagnetization process?

Thank for this question. We did the measurements of both magnets and the magnetisation was the same with the accuracy of 2%. We consider it as the same, taking into account statistic deviation of the properties from sample to sample

(1) The referee is right, the magnetization is not the same. We made changes in the text of supplementary to address this question:

Remanent polarization of sample A is 1.18T, whereas for sample optimally annealed sample B it is 1.197T.

(2) The hysteresis loops of both magnets are different, we describe these differences in the text corresponding to Figures 1 and 2. As for the field of nucleation of a domain of the opposite sign during demagnetization, it is evident from the comparison of the domain structure of both magnets in a field of -0.5 T (Fig 1 and 2) that in both samples the intergranular regions undergo magnetization reversal, while the grains themselves remain magnetized in the initial state. We would not like to resort to the concept of nucleation here, since for such a complex object, this may introduce some confusion into the terminology, since our materials are magnets with pinning mechanism of coercivity, and the nucleation process is not dominant here.

(3) We believe that showing magnetization up to 14 T does not make much sense: in the high fields magnetizations of both samples are saturated and do not change. We think that it is much more important to show the field region where magnetization decreases, therefore, in Fig S1 we show the field region from -3 to +5 T.

(4) In Figures 1 and 2, we have indicated with arrows regions where the first domains appear in the boundary layer.

This was hence added to the caption:

Arrows in some of the images indicate regions where demagnetization initiates and first propagates.

(5) We did not measure the susceptibility, but it is generally accepted that the slope of the magnetization curve corresponds to the initial susceptibility. The steeper the slope, the higher the susceptibility. We have added a corresponding explanation in the text.

(the magnetic susceptibility was defined as the slope of the magnetization curve).

9. In Figure S2, (1) on line 39, the authors present Fig. c for sample A has an average grain size of 74 μm , whereas the grain size in (sample?) B is 200 μm . However, Table S1 shows that the average grain size of sample A is 200 μm , while sample B's is 220 μm , which is inconsistent between the previous and subsequent descriptions. Furthermore, such results cannot be seen from Figures S2c and f. On the contrary, Figure C has larger grain sizes and fewer grain boundaries. (2) Did the authors notice the difference in magnetic domain structure between Figure S2b and e? Have you noticed the differences in size and distribution of those white particles? Have you noticed the differences in magnetic domain structure at grain boundary positions? (3) Did the authors notice the white line in Figure S2f? (4) Should magnetic domain structure images be provided after applying different demagnetization fields? For example, can you provide the magnetic domain images near the inflection points on the demagnetization curve?

(1) The average grain size of 74 μm refers to magnet C, which was initially was part of the study, but was removed eventually. We thus removed the accidentally left mentioning of magnet C in the supplement and replaced the text as follows:

The microstructures and domain patterns of the samples A and B are directly compared in **Figure S2**. Sample A exhibits regions with finer cells and magnetic domains, located close to the GBs within the grains. When saturating and afterwards demagnetizing the sample, these regions again act as weak spots, referred to as low H_c regions, showing magnetization reversal before any other area is affected (cp. **Fig. S2c**). No such low H_c regions are observed in the sample B. Apart from this difference, the microstructures of both samples are closely similar, with compatible grain sizes of 200 μm for sample A and 220 μm for sample B, see Table S1.

(2) Yes, the domain structure in low and high coercive areas is different, we discuss this matter in the text related to Fig 1 and Fig 2.

(3) The white lines in S2(f) is scratches after polishing, and Kerr microscopy magnifies this contrast.

(4) The figures of magnetic domains at different magnetic fields is shown in Fig 1(e) and Fig2(e) including the field near inflection point (0T and -0.5 T)

10. The data source for Table S1 was not clearly explained. The cell size in Table S1 is close to 300nm, significantly different from the 100nm described in line 73. What is the reason for this?

The cell size was measured based on BF-TEM images, such as shown in Figs 3a-b and Fig. 4a. The cell size of 100nm, mentioned in the introduction in line 73, refers to the typical *length scale*. The actual cell size depends on the thermal history and can deviate from this value, as it is true for the magnets in this study.

11. Does the smaller the cell size, the smaller the coercivity? Can the composition measured by SEM-EDS in Table S2 genuinely reflect the composition of the cellular grains for such a small size? What does the precipitate in Table S2 refer to? What is its size?

In our particular case, in the region with lower coercivity, the characteristic size of the cellular microstructure is smaller. This is an experimental fact, although somewhat contrary to the intuitive understanding that the smaller the cell, the greater the coercivity. In the paper, we explain the difference in coercivity not by a slight change in cell size, but by a slight change in the concentration of copper and iron in the 1:5 and 2:17 phases and the appearance of a copper coating on Z-phase.

Precipitate in Table S2 refers to white appearing precipitates with the length scale 10 μm found at the GBs, that appear in both samples, as can be seen in Figs. S2a and S2d. But since they are found similarly in both samples and don't show influence on (de-)magnetization in Kerr images, we omitted the discussion and deleted them from Table S2. As already mentioned, with the exception of the low-coercivity regions of magnet A, the microstructure of the remaining parts of the magnets is completely identical.

12. Does Fig. S4 show that the smaller the magnetic domain size, the lower the coercivity?

Yes, it is true: the lower coercivity areas has finer magnetic domains structure. The physics of this process is quite complex, and an explanation of this phenomenon is beyond the scope

of this article. Here we will only emphasize that the width of the domains is different in different regions, which allows us to identify these regions.

13. Are Figures 1c-d SEM images instead of Kerr microscopy images? Why can't we see magnetic domains or magnetic domain walls? Can you provide magnetic domain images under different demagnetization fields (such as -0.3T, -1.0T) to prove that dark regions are demagnetized first? Can you accurately provide the nucleation field for demagnetizing these dark grains?

Figures 1c-d and 2c-d are Kerr microscopy images.

We would like to refer the reviewer to fig 1e as well as Fig 2e in which we show the domain structure in negative fields starting from approx. -0.3 – -0.5 T, the early stage of the demagnetization is marked with arrows.

14. Comparing Figure 1 and Figure 2, it can be found that there are many defects in sample A, including black particles in the sample and large black particles on the grain boundaries and inside the grains, and the microstructure and magnetic domain structure inside different grains are also different. Does this mean there are significant differences in the two samples' intrinsic properties, such as composition and microstructure?

Figures 1 and 2 show only a small part of the sample, and it is possible that some differences can be noticed here. However, we have carefully examined these samples, collecting statistics from dozens of images, and we can confidently say that in the scanning electron microscope the structure looks very similar. We have also provided all the numerical characteristics of the structure in tables S2 and S3, so that the reader can reliably compare the parameters of the microstructure and the chemical composition of the phases.

15. According to Figure 1 and Figure S2, the low coercivity region mentioned by the authors should be located near the grain boundary, which is several tens of micrometers. However, Figure 3a has a size of less than 2 μm . How did the authors distinguish between low and high coercivity regions in TEM? Is it based on the cell size? Figure 1 shows regions with different cell sizes should be within the same large grain.

The TEM lamella was lifted from the same region that we identified by SEM and Kerr microscopy.

We added the following text to the manuscript

For transmission electron microscopy and atom probe tomography, samples were extracted from areas of both types that had previously been identified by SEM and Kerr microscopy.

16. The authors define different regions in sample A as low-coercivity regions and high-coercivity regions. Still, the highest coercivity of the magnet is 2.2T, so the high coercivity regions in sample A are also defective regions.

This represents the central finding of our work: although the high-coercivity regions in magnets A and B, as well as their grain boundary regions, exhibit very similar compositions,

the presence of localized low-coercivity regions leads to a substantial reduction in the overall coercivity of the material. Using micromagnetic modeling, which confirms our experimental observations, we demonstrated that these low-coercivity intragranular regions act as the critical weak link. Their elimination is therefore essential for further enhancing the performance of industrial Sm–Co-based magnets.

We have changed the text in the conclusion:

Since the high-coercivity regions in magnets A and B have almost the same composition, it is the presence of low-coercivity regions that is the cause of the overall decrease in the coercivity of the entire sample A, which is also fully confirmed by the results of the micromagnetic analysis.

17. Is the density of Z-Platelets in Figure 4a higher than in Figure 3? Is the morphology of each Z-Platelet group also different from that in Figure 3?

Line 228 and 229 can be changed as follows:

228 It shows that 2:17, 1:5 and Z-platelets ...

229 are well developed, although the density of both phases are slightly lower than their density in sample B, but no considerable differences are observed between sample B and reference sample.

18. On line 254, APT confirms the observations by SEM in Fig. 1: the low-Hc region is richer in Sm and Cu by approximately 1 at.% and poorer in Fe and Co by 1–2 at.% compared to the high-Hc region. If there were only such a slight difference in composition, would the anisotropic fields of these phases and the pinning resistance between them differ so much?

The difference is in the microstructure, not in the anisotropic fields. Moreover, the minor composition differences reported here are averaged over the full volume the low Hc and high-Hc regions, but the compositions of the individual phases on the nanoscale differ more substantially, which changes their pinning behavior, as described in the main text.

19. The infocus image in Fig. S7a shows an approximately 1 μm wide region between the high-Hc and low-Hc grain. Why can't such a wide grain boundary have an adverse effect on the coercivity observed in sample A? Why is poor performance necessarily due to the cellular structure within the grains? From Figure S8, it can be seen that the composition within the grain boundaries is also uneven. For example, there are regions rich in Sm-Cu-Zr and poor in Co-Fe in the grain boundaries. Why should these regions not have a low nucleation field? The magnetization curve (Figure 9a) reveals easy demagnetization in the GB region.

In our experimental and theoretical analyses we identified three characteristic regions: (1) high-Hc, (2) low-Hc and (3) transition region (i.e. grain boundary). From these regions, (3) is the soft region without (or low density) pinning sites where the domain walls travel through quickly as a result having low coercivity; (2) low-Hc region with already cellular structure and P1 and P2 pinning sites, but only slightly higher coercivity than (3) (see Figure 9a). Amount of (3) is also relatively low as it is only occupies regions close to grain boundaries and large majority of the grains is the well-developed cellular structure. We observe the effect of region (3) as a small kink in low negative field in the hysteresis loop (Fig. S1).

Most importantly, the (1) high-Hc region contains high strength pinning sites at intersections of 1:5-2:17 (P1) and coated Z-1:5, which is mainly responsible for the high coercivity. These results are described in lines 360-400, 434-440 and in the Abstract.

The grain boundary region indeed has an adverse effect as it is expected to be softer than the inner region and has low density or no pinning sites. The micromagnetic simulations confirmed it. However, improving the micromagnetic simulations is expected to be difficult as the magnetic parameters of the Co-Fe-poor regions are unknown.

20. There are many errors in the references.

We carefully checked references and change the misprints.

21. The writing format also does not meet the journal's requirements.

Thank you for the comment. The manuscript format will be adjusted to the recommendations of the editorial teams of Nature Communications.

Dear Editor,

We would like to thank you and the reviewers for the careful evaluation of our manuscript and for the constructive remarks that helped us to improve the quality and clarity of the work. We have carefully considered all the comments and revised the manuscript accordingly. Below, we provide a detailed response to each point raised by the reviewers, indicating the changes made in the text (in blue) and, where necessary, providing additional explanations to clarify our results and conclusions.

Reviewer #3 (Remarks to the Author):

The reviewer has answered my concerns well, but there are still some issues with the standardization of the writing, such as:

1. The number $\text{Sm}_2(\text{Co}, \text{Fe}, \text{Cu}, \text{Zr})_{17}$ in the title should be subscripted.

This has been modified.

2. The chemical formula writing is confusing, with examples such as $\text{Sm}(\text{Co}, \text{Fe}, \text{Cu}, \text{Zr})_7$ and $\text{Sm}(\text{CoFeCuZr})_7$ appearing throughout the text, some with commas and some without.

This has been homogenised across the manuscript.

3. On page 16, there is still an error (Error! Bookmard not defined) that has not been corrected.

This has been modified.

4. The writing of supplementary materials is very non-standard, such as: (1) The font size in the supplementary materials is inconsistent. (2) The font size in the supplementary materials is inconsistent, such as in Table S3. (3) The tables in the supplementary materials are not standardized. (4) The formula writing and interrupt sign writing in the supplementary materials are not standardized.

Wherever possible, the fonts have been made homogenous and the formatting adjusted.